# Early life stressful experiences escalate aggressive behavior in adulthood via changes in transthyretin expression and function

Rohit Singh Rawat[1], Aksheev Bhambri[1,2], Muneesh Pal[1], Avishek Roy[3], Suman Jain[3], Beena Pillai[1,2], Arpita Konar[1,4]*

[1]CSIR-Institute of Genomics & Integrative Biology, New Delhi, India; [2]Academy of Scientific and Innovative Research (AcSIR), New Delhi, India; [3]All India Institute of Medical Sciences, New Delhi, India; [4]Institute of Health Sciences, Presidency University, Kolkata, India

*For correspondence:
konar.arpita24@gmail.com

Competing interest: The authors declare that no competing interests exist.

**Abstract** Escalated and inappropriate levels of aggressive behavior referred to as pathological in psychiatry can lead to violent outcomes with detrimental impact on health and society. Early life stressful experiences might increase the risk of developing pathological aggressive behavior in adulthood, though molecular mechanisms remain elusive. Here, we provide prefrontal cortex and hypothalamus specific transcriptome profiles of peripubertal stress (PPS) exposed Balb/c adult male mice exhibiting escalated aggression and adult female mice resilient to such aberrant behavioral responses. We identify transthyretin (TTR), a well known thyroid hormone transporter, as a key regulator of PPS induced escalated aggressive behavior in males. Brain-region-specific long-term changes in *Ttr* gene expression and thyroid hormone (TH) availability were evident in PPS induced escalated aggressive male mice, circulating TH being unaltered. *Ttr* promoter methylation marks were also altered being hypermethylated in hypothalamus and hypomethylated in prefrontal cortex corroborating with its expression pattern. Further, *Ttr* knockdown in hypothalamus resulted in escalated aggressive behavior in males without PPS and also reduced TH levels and expression of TH-responsive genes (*Nrgn*, *Trh*, and *Hr*). Escalated aggressive behavior along with reduced *Ttr* gene expression and TH levels in hypothalamus was also evident in next generation F1 male progenies. Our findings reveal that stressful experiences during puberty might trigger lasting escalated aggression by modulating TTR expression in brain. TTR can serve as a potential target in reversal of escalated aggression and related psychopathologies.

## Editor's evaluation

The adolescent phase of life, particularly that surrounding puberty, is a sensitive period for brain development such that adverse experiences during that time have enduring negative impacts but the mechanisms for how this occurs are largely unknown. This important study provides convincing evidence for an unexpected role for the thyroid hormone transporter, transthyretin, which shows region-specific changes in expression following peri-pubertal stress and increased aggression in males, but not females. Mimicking the changes in transthyretin expression induced by stress also increases aggressive behavior in adult males, suggesting a causal connection between changes in thyroid hormone signaling and the behavioral changes induced by stress around puberty.

## Introduction

Escalated aggressive behavior with pathological signatures poses immense risk for violent and anti-social behavior and is a major challenge to human welfare (*Miczek et al., 2007*). Aggression is a behavioral adaptation to threat and competition, but when it becomes excessive, uncontrollable and out of contextd it is considered as maladaptive and pathological (*Neves and Tudela, 2015*; *Waltes et al., 2016*).

Animal aggression is considered escalated and pathological if it displays short attack latency, prolonged attack duration, attacks targeted on inappropriate partners, and body parts prone to serious injury; attacks not signaled by threats; or ignorance of signals of opponents (*Bacq et al., 2020*). In general, animal models of escalated aggression can largely explain abnormal aggressive behavior in humans (*Miczek et al., 2015*). Therefore, it is extremely important to understand the biological factors contributing to shift of normal adaptive aggression to escalated and pathological form.

Mounting epidemiological evidences link early life stressful experiences with deteriorating mental health (*Nelson and Trainor, 2007*; *Duke et al., 2010*; *Haller et al., 2014*; *Hunt et al., 2019*; *Mitchell and Aamodt, 2005*). In particular, stress exposures around puberty including childhood and adolescence including fear, maltreatment, physical and sexual abuse confers susceptibility to aggressive behavioral disorders (*Veenema, 2009*; *Tzanoulinou and Sandi, 2017*; *Bounoua et al., 2020*). Although, early life adversities are considered as one of the potential triggers for abnormal aggression, biological insights are obscure. More importantly, majority of research in the field of aggressive biology have focused on the adaptive form without really considering the escalated and pathological forms (*de Boer, 2018*). *Márquez et al., 2013* developed a novel animal model which showed the effect of peripubertal fearful exposures on male pathological aggression at adulthood in Wistar Han rats. They primarily focused on neural circuits of aggression and on a single gene *Maoa* in isolation.

Considering aggressive behavior as a multidimensional trait, we rationalized that unbiased genome wide investigation would decipher key molecular pathways that can be exploited further as prediction and intervention targets. We modeled peripubertal stress (PPS) induced escalated aggression in laboratory bred Balb/c mice and screened the male cohort showing extremes of behavioral changes. Female mice showed resilience towards PPS induced escalated aggressive behavior as also reported previously (*Konar et al., 2019*).

Next, we performed a sex-specific transcriptome analysis in vulnerable brain regions of hypothalamus (Hypo) and prefrontal cortex (PFC). Hypothalamus is a key brain region for expression of aggressive behavior and neural circuit-specific manipulation experiments revealed that ventromedial hypothalamus is the crucial for inter-male aggression (*Lin et al., 2011*; *Falkner et al., 2016*). While Hypo is considered as the trigger center for aggression, PFC plays opposite regulatory role being involved in inhibition of threat provoked aggressive behavior. More importantly, direct neuronal projections from PFC to Hypo have been suggested to control both type and amplitude of aggressive behavior (*Choy et al., 2018*; *Biro et al., 2018*). Therefore, we primarily focused on hypothalamic molecular culprits of escalated aggression and also included PFC in our study to understand inter-brain regional molecular regulation if any.

We prioritized *Ttr* gene given its (i) top rank in Hypo transcriptome analysis and unique sex specific diametrically opposite expression pattern in Hypo and PFC and (iii) long-term gene expression changes from early peripubertal age till adulthood. Brain-region-specific changes in *Ttr* gene expression and promoter methylation and thyroid hormone (TH) availability, one of the key functions of TTR were evident in PPS-induced escalated aggressive male mice. Further, targeted gene manipulation revealed causal role for hypothalamic *Ttr* in development of escalated aggressive behavior. However, causal relationship of *Ttr* regulated hypothalamic TH availability and escalated aggression is still to be explored.

## Results
### Screening of escalated aggressive behavior

Adult animals exhibiting escalated aggressive behavior in response to PPS exposure were screened based on resident intruder (RI) behavioral scoring parameters. Behavioral scoring data are presented

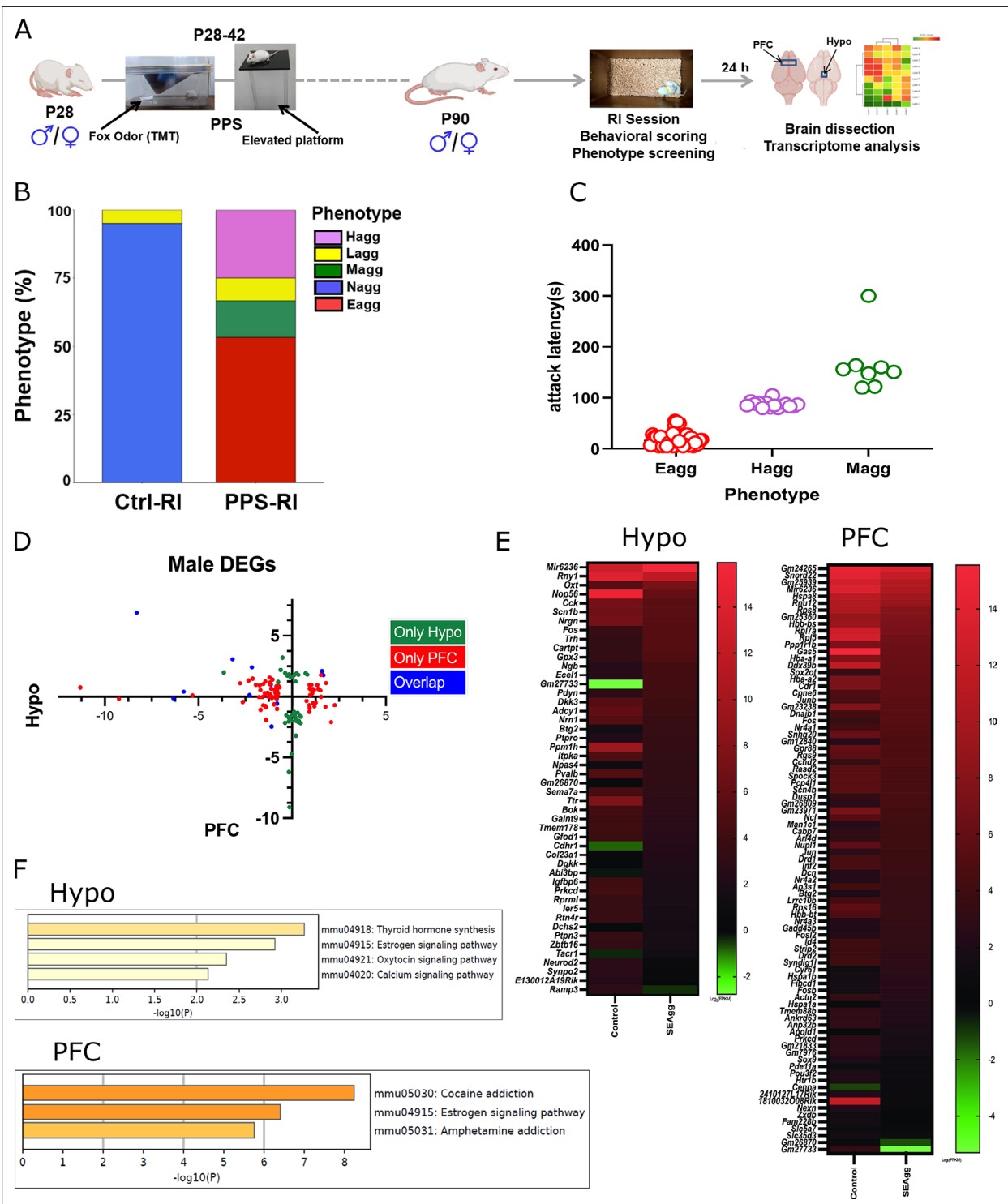

**Figure 1.** Brain-region-specific transcriptional responses in peripubertal stress induced adult males showing escalated aggression. (**A**) Experimental timeline of peripubertal stress (PPS) exposure, resident intruder (RI) behavioral paradigm, brain dissection and transcriptome analysis. (**B**) Phenotypic behavioral screening post RI scoring in control mice without PPS exposure (Ctrl-RI; N=60) and experimental mice with PPS exposure (PPS-RI; N=60) Histogram represents non-aggressive (Nagg; N=57) and less aggressive (Lagg; N=3) mice in the Ctrl-RI cohort. PPS-RI cohort comprises of escalated aggressive (Eagg; N=32), hyper-aggressive (Hagg; N=15), moderate-aggressive (Magg; N=8) and less-aggressive (Lagg; N=5) mice (**C**) Attack latency of Eagg, Hagg and Magg mice of PPS-RI cohort. (**D**) X-Y plot depicting the log 2 fold changes of differentially expressed genes (DEGs) in prefrontal cortex (PFC) in X-axis and hypothalamus (Hypo) in Y-axis and their overlap. (**E**) Heatmap of DEGs in Ctrl-RI (Control) vs PPS-RI escalated aggressive (SEagg) males Hypo and PFC and (**F**) KEGG gene enrichment analysis. RNA sequencing libraries were prepared from N=3 mice/biological replicates per group.

*Figure 1 continued on next page*

*Figure 1 continued*

The online version of this article includes the following source data and figure supplement(s) for figure 1:

**Source data 1.** Data points for attack latency.

**Figure supplement 1.** Volcano plots of differentially expressed genes in brain regions of male mice.

**Figure supplement 2.** Neuron enriched top ranking differentially expressed genes (DEGs) in male hypothalamus (Hypo) determined by Barres Lab data base (https://www.brainrnaseq.org/) for cell-type-specific analysis of FPKM levels of genes in mouse brain.

**Figure supplement 3.** Microglia enriched top ranking differentially expressed genes (DEGs) in male hypothalamus (Hypo) determined by Barres Lab data base (https://www.brainrnaseq.org/) for cell-type-specific analysis of FPKM levels of genes in mouse brain.

**Figure supplement 4.** Non-neuronal cell enriched top ranking differentially expressed genes in hypothalamus of male mice.

**Figure supplement 5.** Neuron enriched top ranking differentially expressed genes (DEGs) in male prefrontal cortex (PFC) determined by Barres Lab data base (https://www.brainrnaseq.org/) for cell-type-specific analysis of FPKM levels of genes in mouse brain.

**Figure supplement 6.** Astrocyte enriched top ranking differentially expressed genes (DEGs) in male prefrontal cortex (PFC) determined by Barres Lab data base (https://www.brainrnaseq.org/) for cell-type-specific analysis of FPKM levels of genes in mouse brain.

**Figure supplement 7.** Microglia enriched top ranking differentially expressed genes (DEGs) in male prefrontal cortex (PFC) determined by Barres Lab data base (https://www.brainrnaseq.org/) for cell-type-specific analysis of FPKM levels of genes in mouse brain.

**Figure supplement 8.** Non-neuronal cell enriched top ranking differentially expressed genes in prefrontal cortex of male mice.

**Figure supplement 9.** Heatmap of few top ranking differentially expressed genes (DEGs) in male prefrontal cortex (PFC) along with cell-type-specific markers using Allen Brain Transcriptomic atlas.

**Figure supplement 10.** Heatmap of few top ranking differentially expressed genes (DEGs) in male prefrontal cortex (PFC) along with cell-type-specific markers using Allen Brain Transcriptomic atlas.

---

in *Figure 1*. Screening parameters were optimized based on earlier reports (*Koolhaas et al., 2013*). We observed that in adult control (Ctrl-RI; N=60) Balb/c mice, 95% (N=57) were non-aggressive (Nagg) while 5% (N=3) showed normal offensive aggression (Lagg) but not escalated. Amongst PPS exposed adult male mice (PPS-RI; N=60), 78.33% (N=47) showed escalated aggression with pathological signs characterized by prolonged fighting with short attack latency, attack on females and anesthetized intruder in all the sessions tested. Amongst these 47 mice, 32 (53.33%) showed extremes of behavioral changes with greater than 80% observation time spent in attack, very short attack latency of less than one minute in all the sessions (*Figure 1B and C* and *Figure 1—source data 1*) and those which attacked both females and anaesthetized intruder. These were referred to as escalated aggressive 'Eagg' and rest as hyper-aggressive 'Hagg' (N=15; 25%). We selected 'Eagg' mice cohort for further molecular and cellular analyses. Some mice of the PPS-RI cohort were moderate-aggressive (Magg; N=8; 13.33%) showing signs of pathological form in some days of the RI session and 8.33% (N=5) showed normal offensive aggression across 7 days of 10-min screening sessions. As reported earlier (*Falkner et al., 2016*) females did not show escalated aggression.

## Transcriptome analyses identify brain region-specific gene signatures in PPS-induced adult males showing escalated aggression and resilient females

To discover unbiased molecular correlates of early life stressful experiences induced escalated aggression and its sex differences we used RNA-sequencing to measure all polyA-containing transcripts in Hypo and PFC of adult control and PPS exposed male and female mice. Now we refer to the adult PPS exposed escalated aggressive male mice group as "SEagg" and adult PPS exposed non-aggressive female mice group as "SNagg". Experiment was performed in 3 individual mice considered as biological replicates for all the samples and tissues were collected 24 h after last RI session. Replicate concordance value of one female biological replicate was less than 0.8 and therefore we excluded the sample in further analyses. Heatmaps of differentially expressed genes (DEG)s were constructed from the global transcriptome analysis. A *Supplementary file 1* containing list of DEGs in males and females has been included. In Hypo, 49 genes were differentially expressed amongst which 28 were down-regulated, 20 were up-regulated and 1 was expressed in SEagg males but not

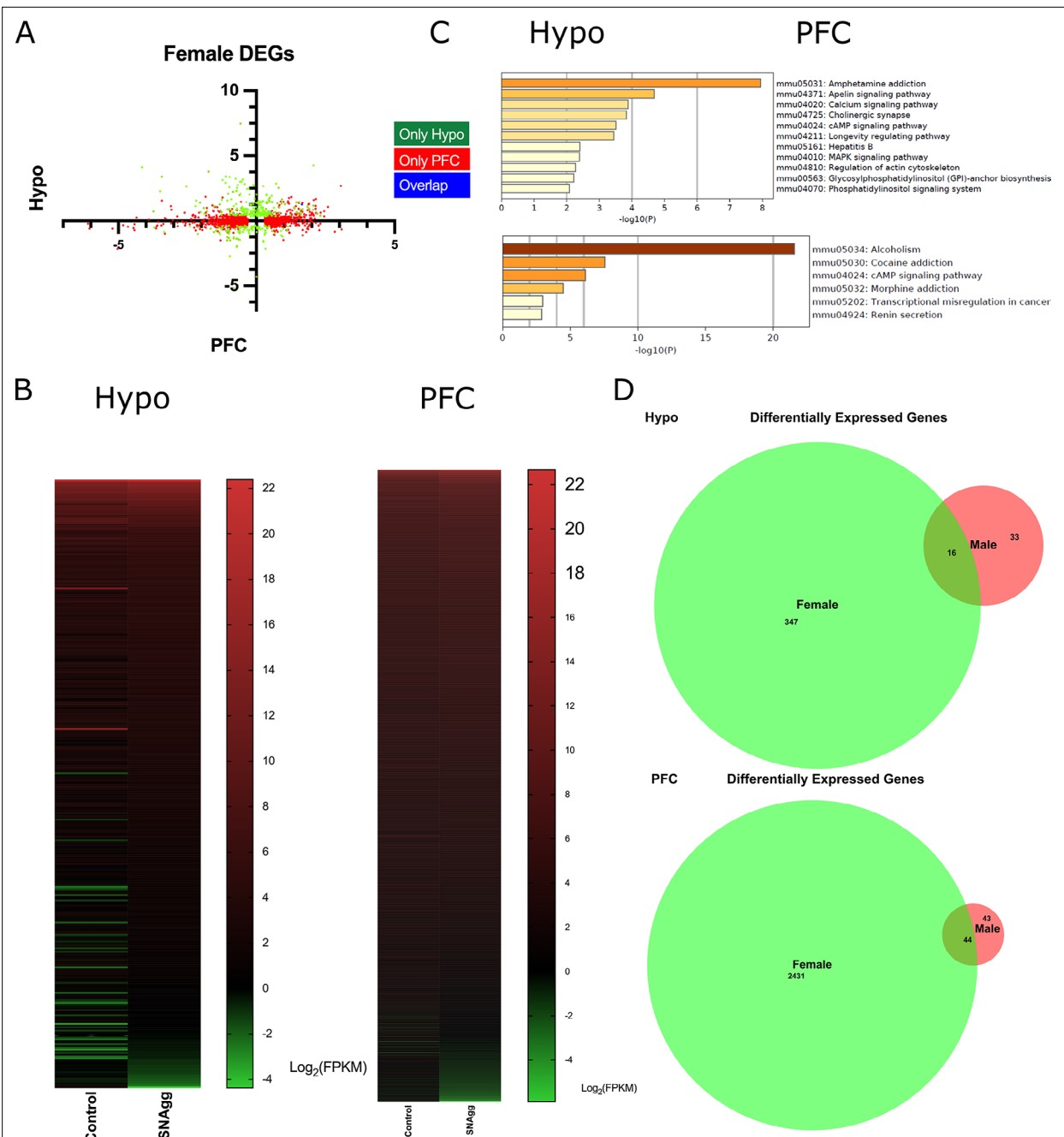

**Figure 2.** Brain-region-specific transcriptional responses in peripubertal stress induced adult resilient females. (**A**) X-Y plot depicting the log 2 fold changes of differentially expressed genes (DEGs) in prefrontal cortex (PFC) in X-axis and hypothalamus (Hypo) in Y-axis and their overlap. (**B**) Heatmap of DEGs in Ctrl-RI (Control) vs PPS-RI Non aggressive (SNAgg) females in Hypo and PFC and (**C**) KEGG analysis and (**D**) Venn diagram of Hypo and PFC specific overlapping DEGs between SEagg males and SNAgg females. RNA sequencing analyses were performed on N=2 mice/biological replicates per group.

The online version of this article includes the following figure supplement(s) for figure 2:

**Figure supplement 1.** Volcano plots of differentially expressed genes in brain regions of female mice.

**Figure supplement 2.** Volcano plot showing *Ttr* and other top ranking differentially expressed genes in hypothalamus.

**Figure supplement 3.** RT-PCR validation of differentially expressed genes (set1) in prefrontal cortex (PFC) and hypothalamus (Hypo) of adult PPS-induced escalated aggressive males (SEagg) vs adult non-stressed control males (Ctrl).

**Figure supplement 4.** RT-PCR validation of differentially expressed genes (set2) in PFC and Hypo of adult PPS induced escalated aggressive males (SEagg) vs adult non-stressed control males (Ctrl).

in control males. PFC of SEagg males showed 87 DEGs amongst which 57 were downregulated, 28 were upregulated and 2 were only expressed in SEagg males but not control males (*Figure 1D and E* and, *Figure 1—figure supplement 1*). Cell type analysis (https://www.brainrnaseq.org/) of top ranking DEGs showed highest number of neuron and microglia enriched genes followed by equivalent number of endothelial, astrocytes and oligodendrocytes enriched genes in Hypo. PFC showed highest number of neuron and astrocytes enriched genes followed by equivalent number of microglia, endothelial and oligodendrocytes enriched genes (*Figure 1—figure supplements 2–8*).

Resilient non aggressive females (SNagg) showed more DEGs (Hypo-363; PFC-2475) than SEagg males when compared to their respective control samples (*Figure 2A and B* and *Figure 2—figure supplement 1*) Comparative analysis of male vs female showed both overlapping and discrete gene signatures. In Hypo, 16 DEGs overlapped between male and female, 15 showing expression changes in opposite direction, 1 in similar direction and 33 genes were exclusive to SEagg males (*Figure 2D*). In PFC, 44 DEGs overlapped between male and female, 29 genes showing expression changes in opposite direction and 15 genes in similar direction.

In order to identify the gene signatures causal for PPS-induced male escalated aggression, we prioritized genes of two categories including (i) male exclusive DEGs in Hypo and PFC (ii) DEGs that showed opposite pattern in both sexes. Amongst these DEGs, we selected top ranking 10 genes from each category and finally 20 DEGs got validated by RT-PCR.

*Ttr*, encoding for thyroid hormone (TH) transporter protein was the topmost ranking gene in Hypo of our transcriptome data in males (*Figure 2—figure supplement 2*) that was validated by RT-PCR. Further, it was the only gene showing unique brain region and sex specific diametrically opposite pattern (*Figure 2—figure supplement 2*). Gene ontology enrichment analysis using KEGG tool combined with literature mining also showed TH signaling as one of the top ranking pathways (*Figure 1F*). TH signaling genes *Nrgn* and *Trh* was amongst the top ranking genes in Hypo (*Figure 2—figure supplement 2*).

Amongst rest of the 17 genes (*Figure 2—figure supplements 3 and 4*), 11 were male exclusive but altered either in Hypo (downregulated- *Nrn1*, *Neurod2* and *Zbtb16*; up-regulated- *Cartpt*, *Gm17508* and *Oxt*) or PFC (downregulated- *Gas5*; upregulated- *Cyr61*, *Dcn* and *Man1c1*) or in both brain regions in similar direction (downregulated-*Ddx39b*) but remained unaffected in females (data not shown). 6 genes overlapped with female in opposite direction but were altered either in Hypo (dowregulated-*Rtn4r* and *Pvalb*) or PFC (upregulated-*Sox2ot, Gm12840*) or in similar direction in both Hypo and PFC (upregulated-*Apold1* and *Btg2*). We, therefore, focused on *Ttr* and carried out functional analysis pertaining to TH signaling in our experimental regime.

## PPS incites persistent changes in Ttr gene expression in both sexes but in opposite manner

RT-PCR validation of the transcriptome data revealed unique brain region and sex biased diametrically opposite expression pattern of the only gene Transthyretin (*Ttr*) in adult mice cohort. *Ttr* was also amongst the topmost DEGs based on fold change and p value. In PPS-induced SEagg males, *Ttr* mRNA showed a decrease of 0.23-fold in Hypo and a robust increase of 8-fold in PFC relative to control (Ctrl) males. On the contrary, adult females that did not show aggressive behavior (SNAgg), *Ttr* mRNA expression pattern was opposite to males, being increased in Hypo (13.6-fold) and drastically reduced (0.55-fold) in PFC relative to control counterparts (*Figure 3B*). Three-way ANOVA showed a significant effect for brain region x sex x treatment interaction {F (1, 88)=685.0, p<0.0001}, significant main effect for treatment {F (1, 88)=98.53, p<0.0001}, sex {F (1, 88)=640.3, p<0.0001} and for brain region (F (1, 88)=1913, p<0.0001); Two-way ANOVA also revealed significant main effect of treatment {F (1, 44)=12.89, p=0.0002 in males; F (1, 44)=99.36, p<0.0001 in females} and brain region {F (1, 44)=891.7 in males; F (1, 44)=1026, p<0.0001 in females} in both sexes. In order to understand whether this gene expression changes was persistent from peripubertal age, we analyzed *Ttr* mRNA in brain regions post 24 hr after PPS exposure. The direction of *Ttr* mRNA changes was similar at peripuberty in both the brain regions and sexes although there were minor differences in the extent. PPS caused drastic reduction in Hypo (0.40-fold) and increase in PFC of *Ttr* mRNA expression (22.3-fold) of males. In females, the changes were reverse being upregulated (13.12-fold) in Hypo and reduced (0.51-fold) in PFC of PPS mice relative to unstressed (NS) controls (*Figure 3C*). Three-way ANOVA showed a significant effect for brain region x sex x treatment interaction {F (1, 88)=669.5, p<0.0001},

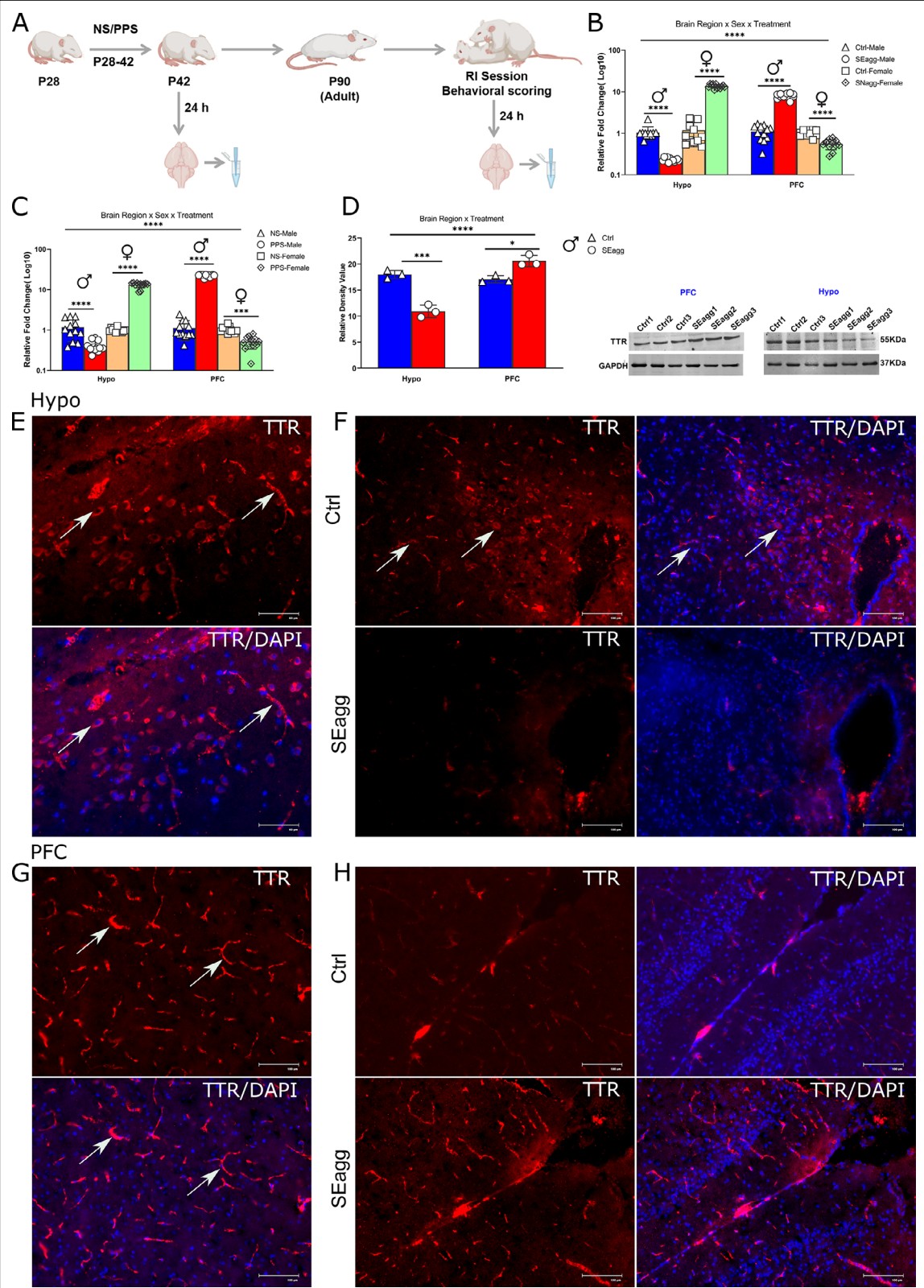

**Figure 3.** Peripubertal stress induced long term changes in TTR expression in brain-region and sex-specific diametrically opposed pattern. (**A**) Experimental timeline for TTR expression analysis. *Ttr* mRNA expression profile in (**B**) Hypo and PFC of peripubertal stress exposed (PPS) adult male (SEagg) and female (SNAgg) mice with control (Ctrl), 24 hr after RI session and (**C**) Hypo and PFC of peripubertal male and female mice 24 hr after stress exposure (PPS) with control [no stress exposure (NS)] counterparts. (N=12 mice/biological replicates per group). Data are presented as mean (±

*Figure 3 continued on next page*

*Figure 3 continued*

SD) and analyzed by three-way ANOVA followed by Bonferroni's post hoc test (**** p<0.0001). (**D**) Representative immunoblot of TTR protein levels with N=3 mice/biological replicates per group. Data are presented as mean (± SD) and analyzed by two-way ANOVA followed by Bonferroni's post hoc test (**** p<0.0001, ***p=0.0001, *p<0.05). Immunofluorescence analysis (N=3 mice/ biological replicates per group) with left panels showing TTR immunoreactive cells (arrowheads) in Hypo (**E**) and PFC (**G**) and right panel showing TTR immunoreactivity in Hypo (**F**) and PFC (**H**) of Ctrl and SEagg mice.

The online version of this article includes the following source data and figure supplement(s) for figure 3:

**Source data 1.** Uncropped immunoblot of TTR and GAPDH.

**Figure supplement 1.** TTR immunofluorescence in mouse brain.

significant main effect for treatment {$F_{(1, 88)}$=196.1, p<0.0001}, sex {$F_{(1, 88)}$=570.1, p<0.0001} and for brain region {($F_{(1, 88)}$=1215, p<0.0001); Two-way ANOVA also revealed significant main effect of treatment {$F_{(1, 44)}$=79.45, p<0.0001 in males; $F_{(1, 44)}$=149.4, p<0.0001 in females} and brain region {$F_{(1, 44)}$=184.6, p<0.0001 in males; $F_{(1, 44)}$=1967, p<0.0001 in females} in both sexes. Details of ANOVA analyses have been given in *Tables 1 and 2* and *Supplementary file 2*.

## TTR protein alters in spatial and cell-type-specific manner

Immunoblot analysis of TTR protein levels corresponded to its transcript pattern in both the sexes. TTR protein was reduced to 0.37-fold in Hypo and upregulated by 1.36-fold in PFC in PPS-induced SEagg males, relative to control (Ctrl) animals (*Figure 3D*). Uncropped immunoblots of TTR has been included (*Figure 3—source data 1*). Two-way ANOVA revealed significant main effect of treatment {$F_{(1, 8)}$=10.17, p=0.0128} and brain region {$F_{(1, 8)}$=60.87, p<0.0001} as well as interaction {F (1,

---

**Table 1.** ANOVA analysis of the results shown in *Figure 3B*.

**A. Three-way ANOVA analysis of the results shown in *Figure 3B*.**

| ANOVA table | SS | DF | MS | F (DFn, DFd) | p value |
|---|---|---|---|---|---|
| Brain Region | 401 | 1 | 401 | $F_{(1, 88)}$=1913 | p<0.0001 |
| Sex | 134.2 | 1 | 134.2 | $F_{(1, 88)}$=640.3 | p<0.0001 |
| Treatment | 20.65 | 1 | 20.65 | $F_{(1, 88)}$=98.53 | p<0.0001 |
| Brain Region x Sex | 5.531 | 1 | 5.531 | $F_{(1, 88)}$=26.39 | p<0.0001 |
| Brain Region x Treatment | 0.2857 | 1 | 0.2857 | $F_{(1, 88)}$=1.363 | p=0.2462 |
| Sex x Treatment | 5.853 | 1 | 5.853 | $F_{(1, 88)}$=27.93 | p<0.0001 |
| Brain Region x Sex x Treatment | 143.6 | 1 | 143.6 | $F_{(1, 88)}$=685.0 | p<0.0001 |
| Residual | 18.44 | 88 | 0.2096 | | |

**B. Two-way ANOVA analysis of the results shown in *Figure 3B* in males.**

| ANOVA table | SS | DF | MS | F (DFn, DFd) | p value |
|---|---|---|---|---|---|
| Interaction | 78.33 | 1 | 78.33 | $F_{(1, 44)}$=447.3 | p<0.0001 |
| Brain Region | 156.2 | 1 | 156.2 | $F_{(1, 44)}$=891.7 | p<0.0001 |
| Treatment | 2.258 | 1 | 2.258 | $F_{(1, 44)}$=12.89 | p=0.0008 |
| Residual | 7.706 | 44 | 0.1751 | | |

**C. Two-way ANOVA analysis of the results shown in *Figure 3B* in females.**

| ANOVA table | SS | DF | MS | F (DFn, DFd) | p value |
|---|---|---|---|---|---|
| Interaction | 65.52 | 1 | 65.52 | $F_{(1, 44)}$=268.5 | p<0.0001 |
| Brain Region | 250.3 | 1 | 250.3 | $F_{(1, 44)}$=1,026 | p<0.0001 |
| Treatment | 24.24 | 1 | 24.24 | $F_{(1, 44)}$=99.36 | p<0.0001 |
| Residual | 10.74 | 44 | 0.244 | | |

**Table 2.** ANOVA analysis of the results shown in *Figure 3C*.

**A. Three-way ANOVA analysis of the results shown in *Figure 3C*.**

| ANOVA table | SS | DF | MS | F (DFn, DFd) | p value |
|---|---|---|---|---|---|
| Brain Region | 305.7 | 1 | 305.7 | F (1, 88)=1,215 | p<0.0001 |
| Sex | 143.5 | 1 | 143.5 | F (1, 88)=570.1 | p<0.0001 |
| Treatment | 49.35 | 1 | 49.35 | F (1, 88)=196.1 | p<0.0001 |
| Brain Region x Sex | 34.57 | 1 | 34.57 | F (1, 88)=137.4 | p<0.0001 |
| Brain Region x Treatment | 1.736 | 1 | 1.736 | F (1, 88)=6.898 | p=0.0102 |
| Sex x Treatment | 0.3444 | 1 | 0.3444 | F (1, 88)=1.368 | p=0.2452 |
| Brain Region x Sex x Treatment | 168.5 | 1 | 168.5 | F (1, 88)=669.5 | p<0.0001 |
| Residual | 22.15 | 88 | 0.2517 | | |

**B. Two-way ANOVA analysis of the results shown in Figure 3C in males.**

| ANOVA table | SS | DF | MS | F (DFn, DFd) | p value |
|---|---|---|---|---|---|
| Interaction | 102.2 | 1 | 102.2 | F (1, 44)=280.4 | p<0.0001 |
| Brain Region | 67.33 | 1 | 67.33 | F (1, 44)=184.6 | p<0.0001 |
| Treatment | 28.97 | 1 | 28.97 | F (1, 44)=79.45 | p<0.0001 |
| Residual | 16.04 | 44 | 0.3646 | | |

**C. Two-way ANOVA analysis of the results shown in *Figure 3C* in females.**

| ANOVA table | SS | DF | MS | F (DFn, DFd) | p value |
|---|---|---|---|---|---|
| Interaction | 68.02 | 1 | 68.02 | F (1, 44)=490.3 | p<0.0001 |
| Brain Region | 272.9 | 1 | 272.9 | F (1, 44)=1967 | p<0.0001 |
| Treatment | 20.72 | 1 | 20.72 | F (1, 44)=149.4 | p<0.0001 |
| Residual | 6.104 | 44 | 0.1387 | | |

8)=90.32, p<0.0001} Details of ANOVA analyses have been given in *Table 3* and *Supplementary file 2*.

Until now we were considering the changes in bulk tissue, therefore, we performed immunofluorescence to elucidate spatial and cell type specificity if any. TTR immunoreactive cells were observed in dorsomedial hypothalamus, ventromedial hypothalamus and arcuate nucleus. Therefore, in later stereotaxy experiments, coordinates were chosen (Materials and methods) to cover all the above mentioned areas. Dorsomedial hypothalamus showed more TTR immunoreactive cells than the other sub-regions. In SEagg males, TTR protein fluorescence intensity was significantly reduced in Hypo (*Figure 3E*) and increased in PFC as compared to Ctrl males. (*Figure 3F*). No primary TTR antibody negative control was used to determine specificity of the immunofluorescence experiments

**Table 3.** Two-way ANOVA analysis of the results shown in *Figure 3D*.

| ANOVA table | SS | DF | MS | F (DFn, DFd) | p value |
|---|---|---|---|---|---|
| Interaction | 84.19 | 1 | 84.19 | F (1, 8)=90.32 | p<0.0001 |
| Brain Region | 56.74 | 1 | 56.74 | F (1, 8)=60.87 | p<0.0001 |
| Treatment | 9.481 | 1 | 9.481 | F (1, 8)=10.17 | p=0.0128 |
| Residual | 7.457 | 8 | 0.9322 | | |

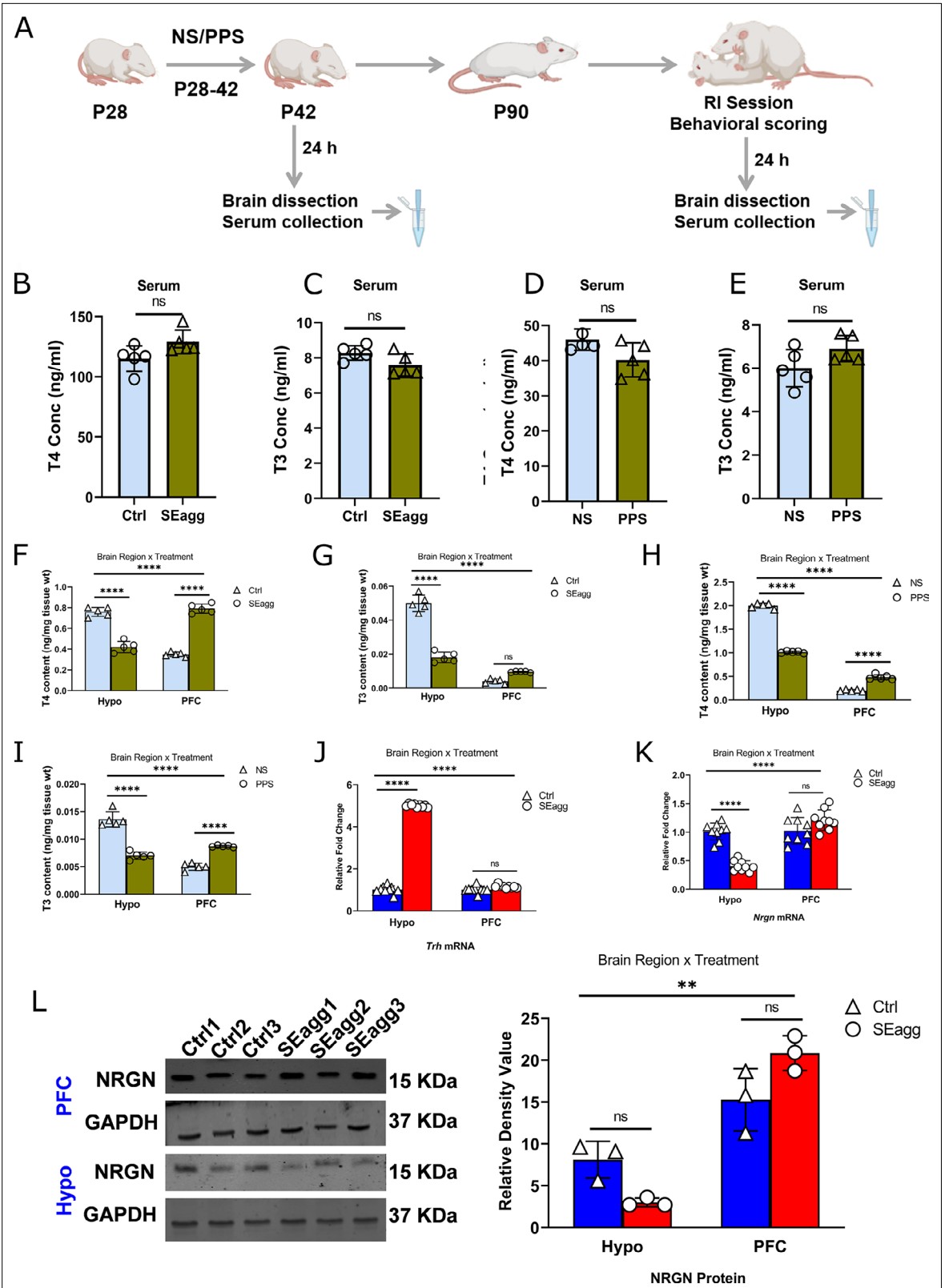

**Figure 4.** Peripubertal stress-induced long-term perturbation of thyroid hormone availability in the brain of escalated aggressive males with concomitant changes in target gene expression. (**A**) Experimental timeline. T4 (**B**) and T3 (**C**) level in serum of peripubertal stress exposed (SEagg) and control (Ctrl) adult males 24 hr after RI session. T4 (**D**) and T3 (**E**) level in serum of peripubertal males 24 hr after stress exposure (PPS) with control [no stress exposure (NS)] counterparts. Data are presented as mean (± SD) and analyzed by unpaired Student's t-test ns (p> 0.05). T4 (**F**) and T3 (**G**) level

*Figure 4 continued on next page*

*Figure 4 continued*

in Hypo and PFC of peripubertal stress exposed (SEagg) and control (Ctrl) adult males 24 hr after RI session. T4 (**H**) and T3 (**I**) level in Hypo and PFC of peripubertal males 24 hr after stress exposure (PPS) with control [no stress exposure (NS)] counterparts. All the above ELISA assays 'B-I' were performed in N=5 mice/biological replicates per group. *Trh* mRNA (**J**) and *Nrgn* mRNA (**K**) expression profile in Hypo and PFC of Ctrl and SEagg males (N=9 mice/biological replicates per group). NRGN protein expression (**L**) profile in Hypo and PFC of Ctrl and SEagg males and representative immunoblot with N=3 mice/biological replicates per group. Data are presented as mean (± SD) and analyzed by two-way ANOVA followed by Bonferroni's post hoc test (**** $p<0.0001$, **$p<0.01$ and ns $p>0.05$).

The online version of this article includes the following source data and figure supplement(s) for figure 4:

**Source data 1.** Data points for T4 and T3 ELISA.

**Source data 2.** Uncropped immunoblot of NRGN and GAPDH protein.

**Figure supplement 1.** Thyroid hormone and Nrgn mRNA levels in the serum and brain of peripubertal stress exposed mice.

(*Figure 3—figure supplement 1*). Further studies on co-localization with specific molecular markers are required to identify the cell type of TTR protein in Hypo and PFC.

Next we investigated TTR protein fluorescence intensity in choroid plexus region, considered to be the main site of TTR protein synthesis in brain, though it did not show any difference between Ctrl and SEagg group. (*Figure 3—figure supplement 1*).

## Long term perturbation of thyroid hormone availability and target gene expression in brain

To explore the functional consequences of perturbed TTR expression, we measured peripheral as well as brain-region-specific T4 and T3 content in both sexes. Circulating TH including total T4 and T3 in serum (*Figure 4B-E* and *Figure 4—source data 1*) was neither altered in adulthood nor at peripubertal age in both sexes. Interestingly, brain TH content was remarkably altered corresponding to *Ttr* gene expression right from peripuberty till adulthood. In adult SEagg males, total T4 and T3 was reduced in Hypo but increased in PFC as compared to control samples (Ctrl) (*Figure 4F and G* and *Figure 4—source data 1*). Two way ANOVA analyses revealed significant interaction between brain region and treatment {$F_{(1, 16)}=408.3$, $p<0.0001$} as well as main effect of treatment {$F_{(1, 16)}=6.711$, $p=0.0197$} though significant main effect of brain region was not observed for changes in T4 content. Ctrl Hypo vs SEagg Hypo and Ctrl PFC vs SEagg PFC groups showed significant differences ($p<0.0001$) in Bonferroni's multiple comparisons test. Two-way ANOVA analyses of T3 content in brain revealed significant main effect of brain region {$F_{(1, 16)}=426.7$, $p<0.0001$}, treatment {$F_{(1, 16)}=100.2$, $p<0.0001$} and interaction {$F_{(1, 16)}=200.5$, $p<0.0001$}. Ctrl Hypo vs SEagg Hypo PFC groups showed significant differences ($p<0.0001$) while Ctrl PFC vs SEagg PFC was not significant in Bonferroni's multiple comparisons test. Of note, statistical analysis by unpaired Student's t-test showed significant difference ($p<0.001$) in Ctrl PFC vs SEagg PFC comparison.

These changes of hypothalamic and PFC T4 and T3 content was persistent from early peripubertal (NS vs PPS males) age (*Figure 4H and I* and *Figure 4—source data 1*). Two way ANOVA analyses of T4 content revealed significant interaction between brain region and treatment {$F_{(1, 16)}=1209$, $p<0.0001$} as well as main effect of treatment {$F_{(1, 16)}=371.9$, $p<0.0001$} and brain region {$F_{(1, 16)}=4097$, $p<0.0001$}. NS Hypo vs PPS Hypo and NS PFC vs PPS PFC groups showed significant differences ($p<0.0001$) in Bonferroni's multiple comparisons test.

Two-way ANOVA analyses of T3 content also revealed significant main effect of brain region {$F_{(1, 16)}=84.87$, $p<0.0001$}, treatment {$F_{(1, 16)}=14.99$, $p=0.0014$} and interaction {$F_{(1, 16)}=189.1$,

**Table 4.** Two-way ANOVA analysis of the results shown in *Figure 4F*.

| ANOVA table | SS | DF | MS | F (DFn, DFd) | p value |
|---|---|---|---|---|---|
| Interaction | 0.7605 | 1 | 0.7605 | $F_{(1, 16)}=408.3$ | $p<0.0001$ |
| Brain Region | 0.002 | 1 | 0.002 | $F_{(1, 16)}=1.074$ | $p=0.3155$ |
| Treatment | 0.0125 | 1 | 0.0125 | $F_{(1, 16)}=6.711$ | $p=0.0197$ |
| Residual | 0.0298 | 16 | 0.001863 | | |

**Table 5.** Two-way ANOVA analysis of the results shown in **Figure 4G**.

| ANOVA table | SS | DF | MS | F (DFn, DFd) | p value |
|---|---|---|---|---|---|
| Interaction | 0.001748 | 1 | 0.001748 | F (1, 16)=200.5 | p<0.0001 |
| Brain Region | 0.003721 | 1 | 0.003721 | F (1, 16)=426.7 | p<0.0001 |
| Treatment | 0.0008738 | 1 | 0.000874 | F (1, 16)=100.2 | p<0.0001 |
| Residual | 0.0001395 | 16 | 8.72E-06 | | |

p<0.0001}. NS Hypo vs PPS Hypo and NS PFC vs PPS PFC showed significant differences (p<0.0001) in Bonferroni's multiple comparisons test.

In females, direction of changes for both T4 and T3 levels were reverse being increased in Hypo and reduced in PFC (*Figure 4—figure supplement 1* and *Figure 4—source data 1*) but remained unaffected in serum.

TH mediates its action by regulating expression of target genes. Therefore, we explored TH responsive genes that was differentially expressed in our transcriptome data (*Trh*, *Nrgn*). Hypothalamic reduction in T4 and T3 content and consequent impaired TH signaling was clearly evident from expression of downstream target genes. TH responsive *Nrgn* mRNA expression showed significant downregulation of 0.5-fold in SEagg males compared to Ctrl males (*Figure 4K*) in Hypo similar to *Ttr* mRNA. Two-way ANOVA analyses showed significant main effect of brain region {F (1, 32)=51.51, p<0.0001}, treatment {F (1, 32)=33.97, p<0.0001} as well as interaction {F (1, 32)=74.27, p<0.0001}. Bonferroni's multiple comparisons test showed significant differences in Ctrl Hypo vs SEagg Hypo (p<0.0001) whereas Ctrl PFC vs SEagg PFC was not significant (p=0.3434). NRGN protein level was also reduced to 0.6-fold in hypothalamus of Eagg males (*Figure 4L*). Two-way ANOVA analyses showed significant main effect of brain region {F (1, 8)=80.63, p<0.0001}, treatment {F (1, 8)=0.02817, p=0.8709} as well as interaction {F (1, 8)=14.71, p=0.0050}. However, Bonferroni's multiple comparisons test did not show significant differences in Ctrl Hypo vs SEagg Hypo (p=0.1916) and Ctrl PFC vs SEagg PFC (p=0.1327) group. Uncropped images of NRGN western blot has been included (*Figure 4—source data 2*). Another, TH regulated gene, *Trh* showed a robust increase of 5-fold in Hypo of SEagg males while remained unaltered in PFC (*Figure 4J*).Two-way ANOVA analyses showed significant main effect of brain region {F (1, 32)=250.7, p<0.0001}, treatment {F (1, 32)=399.6, p<0.0001} as well as interaction {F (1, 32)=282.9, p<0.0001}. Bonferroni's multiple comparisons test showed significant differences in Ctrl Hypo vs SEagg Hypo (p<0.0001), whereas Ctrl PFC vs SEagg PFC was not significant (p=0.1927). Both *Nrgn* and *Trh* mRNA levels showed similar expression profile in early life peripubertal age (*Figure 4—figure supplement 1*) indicating a long term change in gene expression. Details of ANOVA analyses have been given in *Tables 4–10* and *Supplementary file 2*.

## Hypothalamus targeted Ttr knockdown reduced TH levels and induced escalated aggressive behavior in males without peripubertal stress exposure

We checked the direct causal role of TTR by blocking its gene expression through jet-PEI mediated *Ttr* esiRNA injection in hypothalamus. Hypothalamus targeted *Ttr* knockdown to 0.2-fold (80% reduction) in adult unstressed males mirrored the escalated aggression induced by PPS (*Video 1*- Scrambled injected mouse and *Video 2 Ttr* siRNA injected mouse). Escalated aggression in these *Ttr* esiRNA injected male mice was prominent as they showed very short average attack latency of ~15 seconds

**Table 6.** Two-way ANOVA analysis of the results shown in **Figure 4H**.

| ANOVA table | SS | DF | MS | F (DFn, DFd) | p value |
|---|---|---|---|---|---|
| Interaction | 2.003 | 1 | 2.003 | F (1, 16)=1,209 | p<0.0001 |
| Brain Region | 6.786 | 1 | 6.786 | F (1, 16)=4,097 | p<0.0001 |
| Treatment | 0.616 | 1 | 0.616 | F (1, 16)=371.9 | p<0.0001 |
| Residual | 0.0265 | 16 | 0.001657 | | |

**Table 7.** Two-way ANOVA analysis of the results shown in *Figure 4I*.

| ANOVA table | SS | DF | MS | F (DFn, DFd) | p value |
|---|---|---|---|---|---|
| Interaction | 0.000133 | 1 | 0.000133 | F (1, 16)=189.1 | p<0.0001 |
| Brain Region | 5.95E-05 | 1 | 5.95E-05 | F (1, 16)=84.87 | p<0.0001 |
| Treatment | 1.05E-05 | 1 | 1.05E-05 | F (1, 16)=14.99 | p=0.0014 |
| Residual | 1.12E-05 | 16 | 7.01E-07 | | |

and spent 60.5% of total behavioral RI session in clinch attack while none of the scrambled control animals showed signs of attack (*Figure 5B* and *Figure 5—source data 1*). Such behavioral profile was similar to the extent of behavioral changes observed in PPS exposed SEagg male cohort as shown in *Figure 1*.

T3 content in hypothalamus was decreased from 0.07 ng/mg tissue wt in scramble treated group to 0.03 ng/mg tissue wt in *Ttr* esiRNA treated group (*Figure 5C* and *Figure 5—source data 2*). TH-responsive *Trh* mRNA was also markedly increased by 6.5-fold and *Nrgn* mRNA got reduced to 0.36-fold upon *Ttr* gene silencing. Here, we included another well-established TH responsive gene hairless (*Hr*) that showed maximal downregulation to 0.2-fold upon *Ttr* gene silencing in hypothalamus (*Figure 5D*). Chemically induced increase in T4 levels by injecting levothyroxine in PFC did not show any significant behavioral changes (*Figure 5—figure supplement 1*).

## F1 males exhibited escalated aggression, altered Ttr gene expression and TH levels in hypothalamus

We investigated whether PPS-triggered aggression of mouse strain Balb/c is evident in next generation. Adult SEagg males were mated with adult non-stressed females to generate the F1 progenies. F1 male and female progenies were examined at their adulthood. F1 male progenies of SEagg-F0 males showed similar escalated aggressive behavior that characterized the parental generation including short attack latency, attack towards anesthetized and female intruder and now referred to as Eagg-F1. These Eagg-F1 male progenies spent 45% of RI observation time in clinch attack with extremely short attack latency of 11 s while males from control F0 did not exhibit attack (*Figure 6B and C* and *Figure 6—source data 1*). However, female siblings of SEagg-F0 males did not display any prominent sign of aggression.

Next, we checked whether molecular changes in parental generation (SEagg-F0 father) including impaired *Ttr* gene expression and TH levels in brain were also present in the Eagg-F1 males. Similar to SEagg-F0 father, Eagg-F1 males showed deficiency in hypothalamic T3 content while that of PFC was not altered (*Figure 6D* and *Figure 6—source data 2*). *Ttr* expression reduced to 0.35-fold in the Hypo of Eagg-F1 males without any significant change in the PFC. *Nrgn* (reduction to 0.35-fold) and *Trh* (upregulation by 4.3-fold) were also altered similarly in hypothalamus (*Figure 6E*).

## Brain-region-specific DNA methylation changes in Ttr promoter in PPS induced escalated aggressive males

Next, we examined whether epigenetic regulation of *Ttr* could explain the sustained molecular and behavioral changes invoked by PPS exposure. To address this question, we analyzed DNA methylation state of *Ttr* proximal promoter in the Hypo and PFC of SEagg male mice. As anticipated, MedIP qPCR showed that PPS trigger changes in *Ttr* DNA methylation in Hypo and PFC in adulthood. *Ttr* promoter

**Table 8.** Two-way ANOVA analysis of the results shown in *Figure 4J*.

| ANOVA table | SS | DF | MS | F (DFn, DFd) | p value |
|---|---|---|---|---|---|
| Interaction | 10.14 | 1 | 10.14 | F (1, 32)=282.9 | p<0.0001 |
| Brain Region | 8.99 | 1 | 8.99 | F (1, 32)=250.7 | p<0.0001 |
| Treatment | 14.33 | 1 | 14.33 | F (1, 32)=399.6 | p<0.0001 |
| Residual | 1.147 | 32 | 0.03586 | | |

**Table 9.** Two-way ANOVA analysis of the results shown in *Figure 4K*.

| ANOVA table | SS | DF | MS | F (DFn, DFd) | p value |
|---|---|---|---|---|---|
| Interaction | 5.722 | 1 | 5.722 | F (1, 32)=74.27 | p<0.0001 |
| Brain Region | 3.969 | 1 | 3.969 | F (1, 32)=51.51 | p<0.0001 |
| Treatment | 2.617 | 1 | 2.617 | F (1, 32)=33.97 | p<0.0001 |
| Residual | 2.465 | 32 | 0.07705 | | |

showed brain-region-specific differential methylation state in opposite direction to that of its expression pattern. 5-methylcytosine fold enrichment analyses showed increase in Hypo (9.79-fold) and reduction (0.28-fold) in PFC in SEagg males relative to control (Ctrl) (*Figure 7* and *Figure 7—source data 1*). Two-way ANOVA revealed significant main effect of treatment {F (1, 20)=47.97, p<0.0001} and brain region {F (1, 20)=29.68, p<0.0001} as well as interaction between two {F (1, 20)=321.7, p<0.0001 Details of ANOVA analyses have been given in *Table 11* and *Supplementary file 2*.

## Discussion

In the present study, we fill in the existent gap of knowledge about the molecular roots of escalated aggressive behavior. Based on unbiased transcriptome screening, we identify novel role of TTR in long-term programming of escalated aggression induced by PPS. Our findings also indicate the possible involvement of TTR dependent brain TH availability in such abnormal behavioral response. However, further studies are necessitated to identify the definite molecular mechanism.

TTR is a 55 kDa protein that is synthesized in choroid plexus epithelial cells of the brain (*Alshehri et al., 2015*) until recently it was identified in neurons (*Li et al., 2011*; *Zawiślak et al., 2017*) and astrocytes indicating wide expression of the protein in CNS. Interestingly, we observed TTR immunoreactivity in different cell types in Hypo and PFC. TTR expression in PFC was similar to brain endothelial cell marker proteins (*Tang et al., 2017*). Our cell type specificity analysis using publicly available databases also revealed highest expression *Ttr* mRNA in endothelial cells followed by neurons, microglia and astrocytes in mouse brain. Additional experiments mainly colocalization with specific markers are required to confirm the cell types.

We observed long term impact of PPS on *Ttr* expression in brain-region and sex-specific diametrically opposite manner. Such a unique pattern of the only top ranking gene from our transcriptome data intrigued us to explore the functional outcomes. TTR is involved in the uptake of T4 from blood to CSF and local distribution of TH in brain (*Alshehri et al., 2015*). TTR has also been assigned other functions including proteolysis of Neuropeptide Y (*Nunes et al., 2006*), neuroprotection and regeneration of damaged neurons (*Sousa et al., 2004*). We focused on circulating and brain TH levels based on multiple reasons. Our transcriptome data showed significant expression change in TH signaling genes primarily *Trh* and *Nrgn* in Hypo. Several other genes in our list (PFC-*Rasd2*, *Fosl2*, *Nr4a3*, *Inf2*, *Arl4d*, *Dcn*, *Pcp4l1*, *Drd2*, *Syndig1l*, *Hspa1a*, *Spock3*; Hypo- *Oxt*, *Cdhr1*, *Col23a1*, *Dgkk*, *Dkk3*, *Cck*, *Ptpro*) showed overlap with literature available on T3 responsive genes in primary cultured neurons (*Gil-Ibáñez et al., 2014*; *Richard and Flamant, 2018*). Also, TH action in developing brain have been considered as critical determinants of multiple neurological deficits (*Préau et al., 2015*).

In clinical settings, TH abnormalities are diagnosed by serum parameters. However, we show that alterations in *Ttr* gene expression paralleled perturbation in T4 and T3 content in brain tissues of Hypo and PFC without affecting the circulating levels of the hormone. Determining the effective concentration

**Table 10.** Two-way ANOVA analysis of the results shown in *Figure 4L*.

| ANOVA table | SS | DF | MS | F (DFn, DFd) | P value |
|---|---|---|---|---|---|
| Interaction | 85.6 | 1 | 85.6 | F (1, 8)=14.71 | P=0.0050 |
| Brain Region | 469.1 | 1 | 469.1 | F (1, 8)=80.63 | P<0.0001 |
| Treatment | 0.1639 | 1 | 0.1639 | F (1, 8)=0.02817 | P=0.8709 |
| Residual | 46.55 | 8 | 5.818 | | |

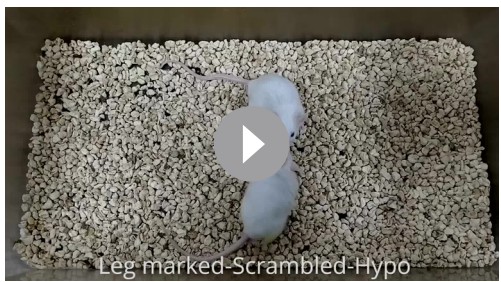

**Video 1.** Video showing resident intruder behavioural session in which resident mouse (legs marked) is scrambled siRNA injected in hypothalamus and intruder is a control mouse of lesser body weight. Scrambled injected mouse did not show any sign of offensive aggression and clinch attack.

https://elifesciences.org/articles/77968/figures#video1

of T4 and T3 in brain tissues is difficult owing to multiple factors driving their synthesis, transport across blood-brain and blood-CSF barrier, intra-cellular distribution and activation/inactivation (*Schroeder and Privalsky, 2014*; *Bárez-López and Guadaño-Ferraz, 2017*). Therefore, future investigations on TH transporters, DIO2 and DIO3 enzymes are necessitated to determine the local TH availability in brain (*Mayerl et al., 2014*; *Williams and Bassett, 2011*).

Any change in brain TH state has a direct influence on TH responsive genes. TH deficiency during postnatal brain development causes irreversible neurological manifestations through target gene expression changes (*Vallortigara et al., 2008*). *Nrgn* is one such brain specific TH responsive gene that was also amongst the top ranking DEGs in our transcriptome data containing TRE elements in promoter and its transcription is dependent on TH in brain (*Pak et al., 2000*; *Husson et al., 2004*). *Nrgn* regulates synaptic plasticity by activating calmodulin kinase II (CaMKII) protein and spine density. We observed significant reduction in *Nrgn* transcript and NRGN protein levels in concordance with decrease in T4 and T3 levels in Hypo.

Until now we found a strong association between altered brain TTR expression and function with escalated aggressive behavior but causal relationship was yet to be established. Next, we showed that intra-hypothalamic *Ttr* gene knockdown evoked escalated aggression in unstressed males to a similar extent to that of PPS induced males. *Ttr* gene knockdown also led to decrease in Hypo T3 content and alteration in expression TH signaling genes, *Nrgn* and *Trh*. *Ttr* gene silencing also reduced hairless (*Hr*), a universal TH responsive gene that is studied to monitor the local TH status in brain (*Herwig et al., 2014*).

Escalated aggressive behavior, reduced *Ttr* mRNA expression and T3 content in Hypo was also evident in F1 male progenies of SEagg males. Previous studies suggest that TH changes in neonatal brain can elicit neuroendocrine abnormalities in their F1 progenies. Also, developmental exposure of thyroxine disrupting chemicals can affect gene expression and behavior in later generations (*Morte et al., 2018*). These data indicated that behavioral and molecular consequences are dependent on TTR. However, it is not clear whether the decrease of TTR expression causes the escalated aggression via the regulation of hypothalamic TH availability or the effect of altered TTR expression on the aggression and TH availability are independent.

*Ttr* promoter showed altered methylation pattern in Hypo and PFC of SEagg males. As DNA methylation is considered important for lasting influence of stressful experiences (*Vukojevic et al., 2014*; *Gulmez Karaca et al., 2020*), it might underlie long-term impact of PPS on *Ttr* expression and aggressive behavior. Further studies are required to determine the molecular pathways underlying involvement of TTR in aggressive behavior.

In conclusion, we delineate novel role of TTR in manifestation of early life stress induced escalated aggression. TTR signaling in brain can also serve as a valid molecular predictor as well as intervention target in excessive pathological aggression. Our findings have inherent limitations of investigations in animal models and therefore further studies are warranted in relevant human cohort to establish role of TTR in abnormal aggression and related psychopathologies. Our work

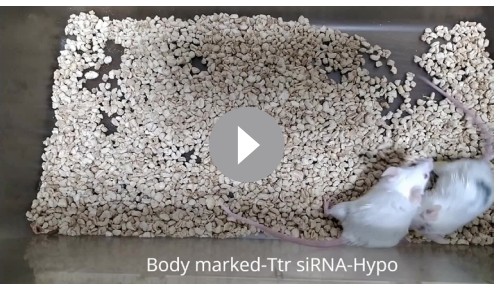

**Video 2.** Video showing resident intruder behavioural session in which resident mouse (body marked) is *Ttr* siRNA injected in hypothalamus and intruder is a control mouse of lesser body weight. *Ttr* siRNA injected mouse showed escalated aggression spending maximum time in clinch attack and biting during the entire behavioural session.

https://elifesciences.org/articles/77968/figures#video2

also provides resource for investigating sexual dimorphism in behavioral disorders and deciphering susceptibility as well as protective pathways.

# Materials and methods

**Key resources table**

| Reagent type (species) or resource | Designation | Source or reference | Identifiers | Additional information |
|---|---|---|---|---|
| Antibody | anti-TTR (rabbit polyclonal) | Thermo Fisher Scientific | Cat# PA5-80196 | WB (1:1000) IF (1:300) |
| Antibody | anti-Nrgn (goat polyclonal) | Abcam | Cat#ab99269 | WB (1:1000) |
| Antibody | anti-GAPDH (mouse monoclonal) | Santa Cruz | Cat#: sc32233 | WB (1:5000) |
| Antibody | anti-5-methyl cytosine (mouse monoclonal) | Epigentek | Cat#: A-1014–050 | MeDIP (1 µg per reaction) |
| Chemical compound, drug | in vivo-jetPEI | Polyplus Transfection | Cat#: 201–10 G | Transfection reagent for in vivo delivery of nucleic acids |
| Chemical compound, drug | 2,4,5-Trimethylthiazole | Sigma-Aldrich | Cat#W332518 | |
| Chemical compound, drug | L-Thyroxine | Sigma-Aldrich | Cat#:T2376-100MG | |
| Commercial assay or kit | EliKine TM Thyroxine (T4) ELISA Kit | Abbkine | Cat#: KET007 | |
| Commercial assay or kit | EliKineTM Triiodothyronine (T3) ELISA Kit | Abbkine | Cat#: KET006 | |
| Sequence-based reagent | *Ttr* esiRNA (esiRNA targeting mouse *Ttr*) | Sigma-Aldrich | Cat#: EMU030721 | |

## Animals

All experimental procedures involving live animals were approved by the Institutional Animal Ethics committee (IAEC) of CSIR-Institute of Genomics and Integrative Biology (IAEC Approval Number-IGIB/IAEC/3/15) that is registered under Committee for the Purpose of Control and Supervision of Experiments on Animals (CPCSEA), Department of Animal Husbandry and Dairying, Ministry of Fisheries, Animal Husbandry and Dairying, Government of India (Registration No and Date- 9/1999/CPCSEA). Male and female offspring of Balb/c mice bred in the institutional animal house were used for the study. All animals were group housed with 3–4 mice per cage under SPF conditions. They were kept in individually ventilated cages (IVC) at 24 ± 2°C on a 12 hr light/dark cycle with ad libitum access to food and water. Animal handling and experiments were conducted in accordance with the institutional guidelines.

## PPS stress procedure

Male and female mice were exposed to unpredictable fear inducing stressors of fox odor {2,3,5-trimethyl-3-thiazoline (TMT) secreted from fox anal gland and component of fox urine and feces} and elevated platform during the peripuberty period of postnatal day (P) 28 to P42 as per the protocol published previously (*Márquez et al., 2013*; *Konar et al., 2019*). Post weaning at P21, equivalent number of mice from different litters were mixed and placed in control (no stress during peripuberty) and experimental (peripubertal stress-PPS) groups in different home cages (3–4 mice per cage) avoiding placing of siblings in the same home cage.

Briefly, P28 male and female offspring were exposed to an open-field for 10 min for acclimatization in a novel environment. Thereafter, one group of mice were exposed to 9 µl of fox odor (Sigma) soaked cloth kept in a filter top plastic cage and elevated platform (96 cm above ground) for 7 random days (P28, P29, P30, P34, P36, P40, and P42) across P28 to P42. Stressors were applied during the active phase of the mice, singly or in combination in variable schedule so that the animals do not learn and get suddenly stressed. The duration of stress session was 25 min following which mice were returned to their home cages. Control animals were handled on the days in which their counterparts were exposed to PPS.

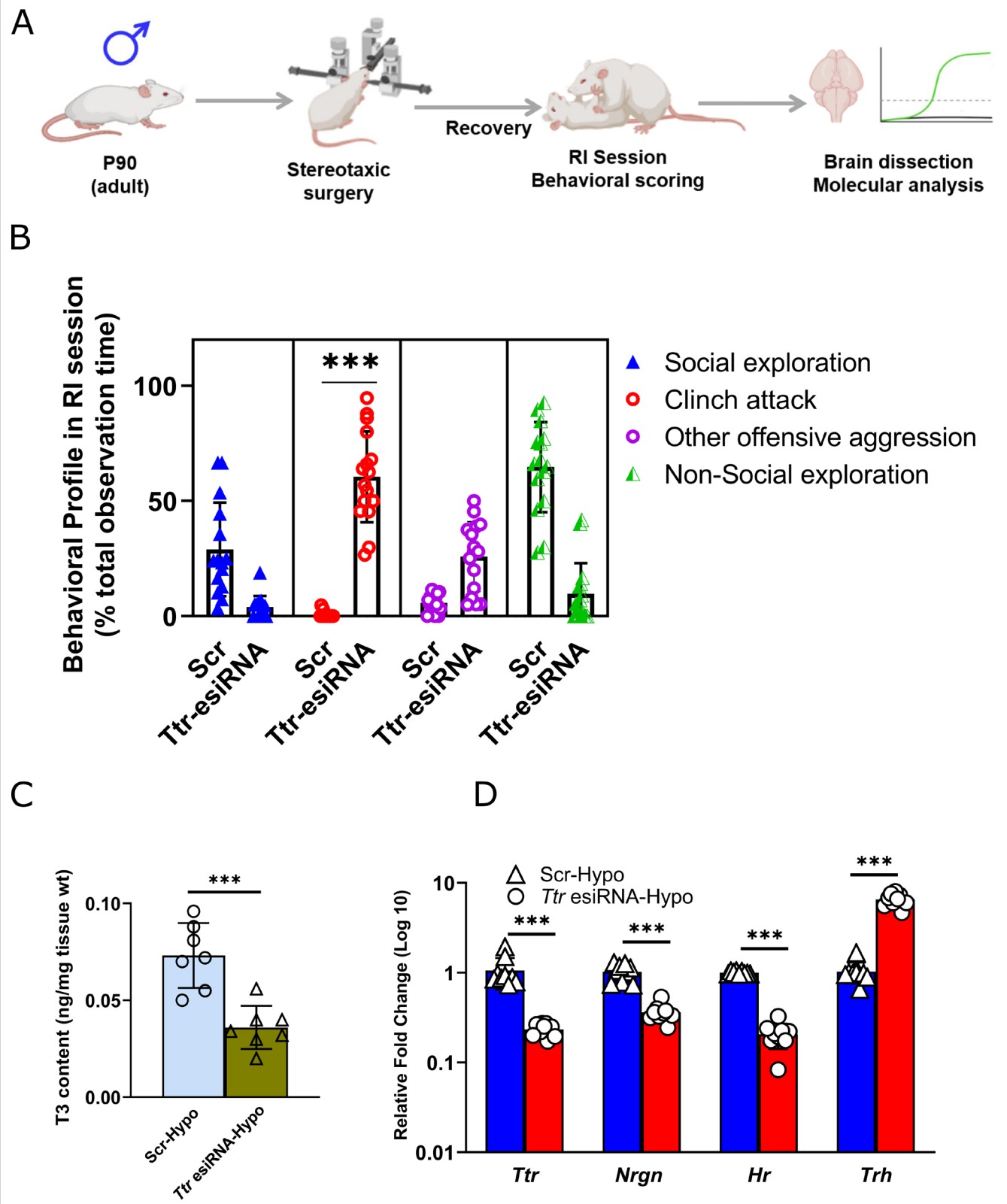

**Figure 5.** *Ttr* knockdown in hypothalamus resulting in escalated aggression and reduced thyroid hormone levels. (**A**) Experimental strategy of stereotaxic surgeries followed by behavioral and molecular experiments (**B**) Comparative analysis of behavioural profile during RI session between hypothalamus injected *Ttr* esiRNA (N=16 mice/biological replicates per group) and scrambled siRNA (Scr) males (N=15 mice/biological replicates per group). (**C**) T3 content in hypothalamus (Hypo) of Ttr esiRNA (N=7 mice per group) and Scr males (N=6 mice/biological replicates per group). (**D**) *Ttr*

*Figure 5 continued on next page*

*Figure 5 continued*

mRNA, *Nrgn* mRNA, *Hr* mRNA and *Trh* mRNA expression analysis in Hypo of *Ttr* esiRNA and Scr males (N=9 mice/ biological replicates per group). Data are presented as mean (± SD) and analysed by unpaired Student's t-test [*** (p< 0.001)] between Scr vs *Ttr* esiRNA group.

The online version of this article includes the following source data and figure supplement(s) for figure 5:

**Source data 1.** Data points for RI behavioral scoring.

**Source data 2.** Data points for T3 ELISA.

**Figure supplement 1.** Comparative analysis of behavioral profile during RI session between saline injected group in PFC (SA-PFC; N=9) and levothyroxine, LT4 injected in PFC (LT4-PFC;N=9).

## Resident intruder (RI) paradigm

RI test for aggression was performed in male and female 'adult control' mice who were not exposed to stress during peripuberty and 'PPS adult' mice who were peripubertally stressed based on the protocol reported earlier (*Márquez et al., 2013*). Animals were individually housed for 1 week prior to testing and RI test was performed in their active phase. Each of these mice referred to as "resident" was exposed to various category of unfamiliar intruders once a day for 10 min for 7 consecutive days. Each day the resident was introduced to a different intruder in the following manner: day 1-same sex and 10% less body weight; day 2-same sex and 10% more body weight; day 3 and day 5-anesthetized of same sex and similar body weight; day4 and day7-opposite sex and similar body weight; day 6-different strain of same sex and similar body weight.

The behavioral parameters including clinch attack, move towards, social exploration, ano-genital sniffing, rearing, lateral threat, upright posture, keep down, chase, non-social explore and rest or inactivity were quantified in terms of percentage (duration) of the total observation time. Attack latency or the time between introduction of the intruder and first clinch attack was also determined. The total duration of the clinch attack, offensive upright, keeping down and lateral threat were considered as the measure of total offensive behavior. Social exploration behavior included the sum of social explore, auto and social grooming and ano-genital sniffing. Phenotypic screening of animals was done based on conventional parameters as published in earlier reports (*Koolhaas et al., 2013*; *Takahashi and Miczek, 2014*) and described in details in result section of *Figure 1*. Briefly, animals showing excessive aggression with pathological signs of very short attack latency, prolonged attack duration, attack on female and anesthetized intruder in all the days of RI test was referred to as escalated aggressive. The order of RI testing for control and PPS exposed adult animals was random. Behavioral scoring was done by an observer blind to animal identity and assignment of groups.

## Breeding scheme for F1offspring

Control (without PPS exposure) and PPS exposed adult male mice showing escalated aggression was mated with control females (without PPS exposure). After mating, males were immediately removed from the cage so that they do not have any contact with their offspring and do not impact upon their rearing. F1 offspring originating from these pairings were housed in standard cages and subjected to RI test for aggression at their adulthood (P90).

## RNA-sequencing

RNA was isolated from hypothalamus and PFC of male and female mice using Trizol reagent. One µg RNA (by Qubit measurement) was taken per sample and RNA sequencing libraries were made using Illumina Truseq Ribo-Gold Total RNA stranded kit as per manufacturer's protocol. The libraries made were quality checked using Qubit and Bioanalyzer. The sequencing was performed using Illumina

**Table 11.** Two-way ANOVA analysis of the results shown in *Figure 7*.

| ANOVA table | SS | DF | MS | F (DFn, DFd) | p value |
|---|---|---|---|---|---|
| Interaction | 8469 | 1 | 8469 | F (1, 20)=321.7 | p<0.0001 |
| Brain Region | 781.5 | 1 | 781.5 | F (1, 20)=29.68 | p<0.0001 |
| Treatment | 1263 | 1 | 1263 | F (1, 20)=47.97 | p<0.0001 |
| Residual | 526.6 | 20 | 26.33 | | |

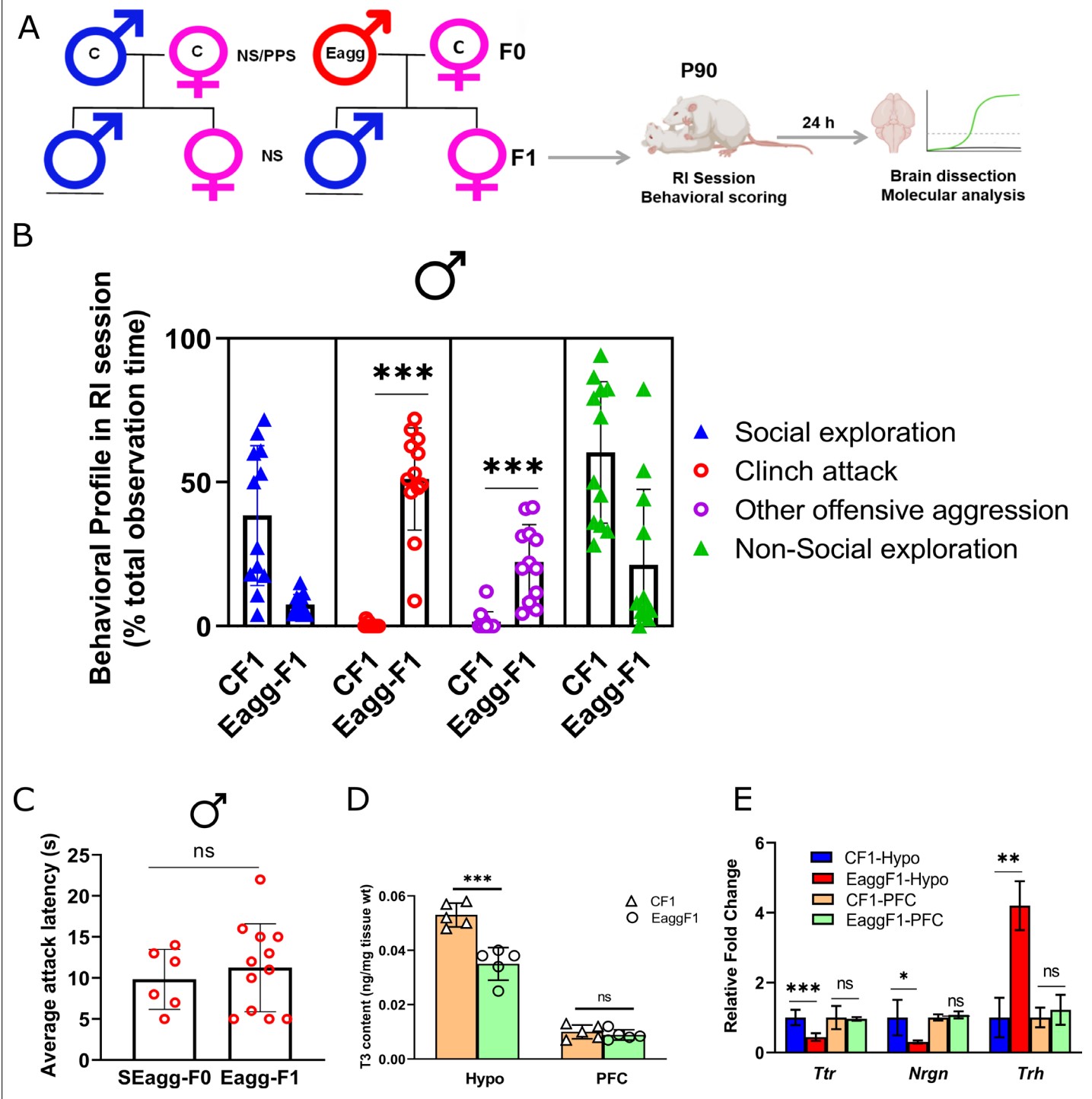

**Figure 6.** F1 male progenies showed escalated aggression with concomitant changes in Ttr gene expression and thyroid hormone signaling. (**A**) Breeding pairs and experimental timeline. (**B**) Comparative analysis of behavioral profile during RI session between F1 males originating from control males crossed with control females (CF1) and F1 males originating from peripubertal stress exposed SEagg-F0 males crossed with control females (Eagg-F1) males (N=12 mice/biological replicates per group). (**C**) Attack latency comparison between parent SEagg-F0 (N=6 mice) and Eagg-F1 males (N=12 mice). (**D**) T3 content in Hypo and PFC of CF1 and Eagg-F1 males (N=5 mice/biological replicates per group). (**E**) *Ttr*, *Nrgn* and *Trh* mRNA expression analysis in Hypo and PFC of CF1 and Eagg-F1 males (N=6 mice/biological replicates per group). Data are presented as mean (± SD) and analyzed by unpaired Student's t-test {ns (p> 0.05),* (p< 0.05), ** (p< 0.01), and *** (*P*<0.001)} between CF1 and Eagg-F1 groups or SEagg-F0 vs EaggF1.

The online version of this article includes the following source data for figure 6:

**Source data 1.** Data points for RI behavioral scoring and attack latency.

*Figure 6 continued on next page*

*Figure 6 continued*

**Source data 2.** Data points for T3 ELISA.

HiSeq 2500 platform in 150cycles paired end format. The raw fastq files were used to check for quality control using FastQC program. Trimmomatic was used to perform trimming using the default paired end parameters in the software. After trimming, quality check was performed again using FastQC and all reads were found to be above phred score 22 with no adapter contamination or over-represented sequences. The fastq files were aligned to Mouse Genome (mm10) using Tophat2 and the bam files were used to calculate differential expression using Cuffdiff2. Due to the stranded nature of the data, fr-firststrand parameter was used during analysis. The FDR corrected pvalue (also called as q-value) was used to check significance of differentially expressed genes. R-studio with ggplot package was used to generate plots while some plots were generated using Prism software. Raw and processed RNA seq datasets including raw transcriptomic data on the transcript levels in each biological replicate of control and experimental samples were deposited in National Center for Biotechnology Information (NCBI) Gene Expression Omnibus (GEO), accession number GSE199844. Replicates that passed the concordance test ($R>0.8$) were included for analysis of differentially expressed genes.

## RT-PCR

Total RNA was isolated from hypothalamus and PFC of mice and 2 µg of RNA from each group was reverse transcribed to cDNA synthesis. RT-PCR was carried out using SYBR Green master mix for detection in Light cycler LC 480 (Roche). All primers used for qRT-PCR are given in *Supplementary file 3*. The endogenous control GAPDH was used to normalize quantification of the mRNA target.

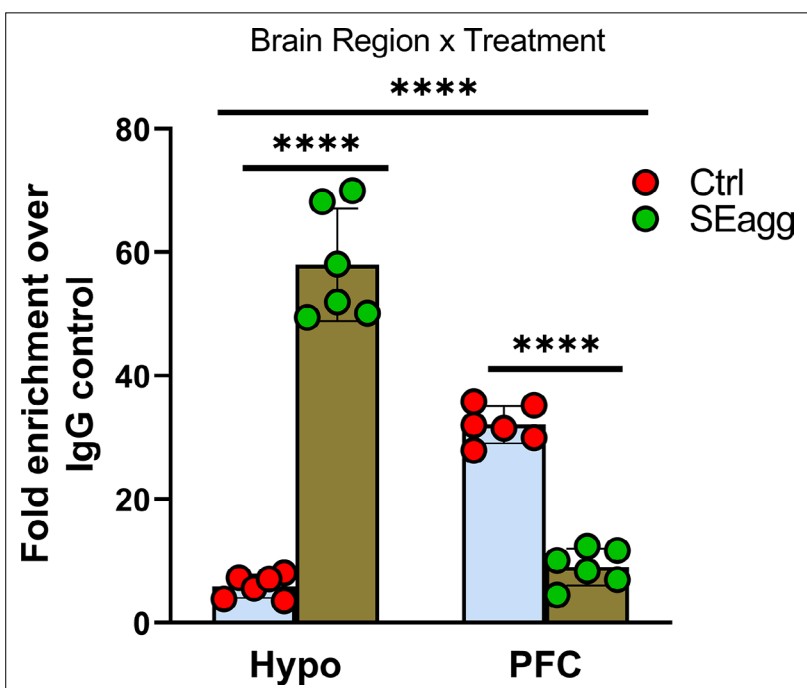

**Figure 7.** *Ttr* promoter methylation changes in PPS-induced escalated aggressive males. Methylated DNA immunoprecipitation analysis showing 5-methylcytosine fold enrichment in *Ttr* promoter in Hypo and PFC of Control male mice (Ctrl) and PPS-induced male mice showing escalated aggression (SEagg) (N=6 mice/biological replicates per group). Data are presented as mean (± SD). Two way ANOVA revealed significant interaction between brain region and treatment (****$p<0.0001$). Bonferroni's multiple comparisons test showed significant difference between Ctrl and SEagg group in both Hypo and PFC region (****$p<0.0001$).

The online version of this article includes the following source data for figure 7:

**Source data 1.** Data points for *Ttr* promoter 5-methylcytosine fold enrichment.

## Immunoblotting

Cytosolic protein fraction was isolated from mouse hypothalamus and PFC, resolved by 10% SDS PAGE and transferred on to PVDF membrane. The membrane was probed with antibodies and target proteins were detected using LiCor Odyssey imaging system. The primary antibodies {anti-TTR rabbit polyclonal, anti-Nrgn rabbit polyclonal; anti-GAPDH mouse monoclonal} and secondary antibodies {anti-rabbit IgG HRP (Cell Signaling Technology, 7074P2) and anti-mouse IgG HRP (Cell Signaling Technology, 7076P2)} were used at adequate dilutions.

## Immunohistochemistry

Mice were anaesthetized with thiopentone (40 mg/kg) and perfused with cold 4% paraformaldehyde in PBS. Brains were removed, post-fixed, cryoprotected in PBS +15% sucrose for 2–3 hr followed by immersion in PBS +30% sucrose for 24 hr, and then sectioned coronally (7 µm) on a cryotome. Free-floating sections were permeabilized with blocking buffer (PBS +3% normal donkey serum, 0.3% Triton X-100) for 2 hours and then incubated with TTR primary antibody overnight at 4 °C. Slices were then washed 4×15 min with PBS, incubated with corresponding secondary antibodies for 2 hr, washed 4×15 min with PBS, mounted on microscope slides followed by counterstaining with DAPI and photomicrographs were captured by FLoid fluorescence microscope.

## Thyroid hormone measurement

Mouse blood samples were collected from heart to test serum levels of total tetraiodothyroxine (T4) and total tri-iodothyroxine (T3). Thyroid hormone content in brain regions was determined by dissecting hypothalamus and PFC and individually homogenizing them in artificial cerebral spinal fluid and centrifuged at 14,000 rpm for 15 min at 4 °C. The resulting supernatant was collected and used for ELISA based determination of total T4 and T3 (EliKineTM Thyroxine (T4) ELISA Kit KET007 and EliKineTM Triiodothyronine (T3) ELISA Kit KET006).

## Stereotaxic surgeries and gene manipulation

Mice were anesthetized with 40 mg/kg BW thiopentone i.p. and positioned on a robotic stereotaxic frame (Cat no. 51700, Stoelting Co., USA) with motorized stereo-drive (Cat no. 013.641, Neurostar, USA). As mentioned in earlier reports (*Hu et al., 2005*), the dorsomedial, ventromedial and arcuate nucleus of hypothalamus were targeted bilaterally by using the stereotaxic coordinates of 1.5 mm posterior to the bregma, 0.5 mm lateral to midline, and 5.8 mm below the surface of the skull. For PFC, the specific coordinates for injection relative to bregma was mediolateral ±0.35 mm, dorso-ventral –2.1 mm, and rostrocaudal axes +2.2 mm. For brain targeted gene manipulation *Ttr* esiRNA (esiRNA targeting mouse *Ttr* - EMU030721, Sigma Aldrich)–jetPEI complex was infused into hypothalamus. Levothyroxine (LT4) was bilaterally administered into PFC. Injection was given using 10 µl Hamilton syringe (Cat no. 72–1823,32 G; 700 N glass) placed in arm-held Elite-11 mini pump (Harvard Apparatus, USA) at a rate of 100 nl/min and the system was left in place for an additional 1 min and then gently withdrawn. Mice were allowed to recover individually from anesthesia and thereafter returned to their home cages. Post-operative cares were taken using analgesics and anti-biotics including meloxicam (5 mg/kg of b/w, i.m., Intas Pharmaceuticals, India) and gentamycin (5 mg/kg bw, i.m., Neon Laboratories, India) for 2–3 days. Body temperature was maintained during and after surgery in homoeothermic monitoring system (Harvard Apparatus, USA) following previous protocol. RI test for aggression was performed followed by molecular experiments after an additional 24 hr.

## Methylated DNA immunoprecipitation

DNA methylation was analyzed at the promoter region of *Ttr* by methylated DNA immunoprecipitation (MeDIP) method as mentioned earlier (*Mukhopadhyay et al., 2008*; *Mohn et al., 2009*). Briefly, 4 µg of sonicated DNA (DNA fragment size ranging from 300 to 1000 bp) isolated from hypothalamus and PFC of Ctrl and SEagg male mice was diluted in immunoprecipitation buffer and incubated with 2 µg 5-methyl cytosine antibody (A-1014; Epigentek) at 4 °C overnight. Mouse IgG Isotype control antibody (02–6502, Thermo Fisher Scientific) was used for mock IP. Next day, 50 µL of Protein A-dynabeads was added and incubated at 4 °C for 2 hr with rotation. Thereafter, it was centrifuged at 3500xg at 4 °C for 10 min and the supernatant was removed carefully. After washing the pellet, the immune complex was eluted, DNA was purified and dissolved in TE buffer. Using eluted DNA as

template, *Ttr* proximal promoter –184 to –33 bp from TSS was amplified with specific primers (**Supplementary file 3**) generating a 151 bp product in MedIP-qPCR.

## Statistical analyses

All statistical analyses were performed using Microsoft Excel or Prism 8 (GraphPad Software). Sample sizes, statistical methods and p values are mentioned in results and figure legends. In order to analyze RT-PCR data, the $2^{-\Delta\Delta Ct}$ value was used to calculate relative fold change in mRNA expression and plotted. For immunoblot analysis, the signal intensity (Integrated Density Value, IDV) of TTR and NRGN bands was measured by spot densitometry tool of AlphaEaseFC software (Alpha Innotech Corp, San Jose, CA, USA), normalized against the IDV of internal control GAPDH and plotted as relative density value. MeDIP data were plotted as fold enrichment normalized to IgG control. Data are presented as mean (± SD) and individual data points are depicted in figure panels, wherever possible. Three-way ANOVA with the factors of sex, brain region, and treatment and two-way ANOVA with the factors of treatment and brain region was used. Within sex or brain region effects were determined by two-way ANOVA. Bonferroni post hoc test was applied for multiple comparisons. Individual comparisons were also made using the unpaired Student's t test. Significance level was set to <0.05.

## Acknowledgements

We acknowledge the animal house facility of CSIR-IGIB, New Delhi, India. We thank Ashish Kumar (Centre for Biomedical Engineering, IIT Delhi, India) for assistance in stereotaxy experiments. Funding: This work was supported by grants from Department of Science and Technology, Govt of India (DST/INSPIRE/04/2014/002261/GAP0125), Department of Biotechnology, Govt of India (GAP0197) and Indian Council of Medical Research (IR-594/2019/RS).

## Additional information

### Funding

| Funder | Grant reference number | Author |
| --- | --- | --- |
| Department of Science and Technology, Ministry of Science and Technology, India | Inspire Faculty Award DST/INSPIRE/04/2014/ 002261 | Arpita Konar |
| Department of Biotechnology, Ministry of Science and Technology, India | Research Grant GAP0197 | Beena Pillai |
| Indian Council of Medical Research | Research Grant IR-594/2019/RS | Beena Pillai |

The funders had no role in study design, data collection and interpretation, or the decision to submit the work for publication.

### Author contributions

Rohit Singh Rawat, Data curation, Formal analysis, Validation, Investigation, Methodology, Writing - original draft; Aksheev Bhambri, Data curation, Software, Formal analysis, Validation, Investigation, Visualization, Methodology; Muneesh Pal, Data curation, Formal analysis, Validation, Investigation, Visualization, Methodology; Avishek Roy, Data curation, Investigation, Methodology; Suman Jain, Resources, Supervision; Beena Pillai, Resources, Funding acquisition, Writing - review and editing; Arpita Konar, Conceptualization, Resources, Data curation, Formal analysis, Supervision, Funding acquisition, Validation, Investigation, Visualization, Methodology, Writing - original draft, Project administration, Writing - review and editing

### Author ORCIDs

Rohit Singh Rawat ![ORCID] http://orcid.org/0000-0001-6329-6636
Avishek Roy ![ORCID] http://orcid.org/0000-0003-2633-487X

Arpita Konar ⬤ http://orcid.org/0000-0002-1761-1065

### Ethics

All experimental procedures involving live animals were approved by the Institutional Animal Ethics committee (IAEC) of CSIR-Institute of Genomics and Integrative Biology (IAEC Approval Number-IGIB/IAEC/3/15) that is registered under Committee for the Purpose of Control and Supervision of Experiments on Animals (CPCSEA), Department of Animal Husbandry and Dairying, Ministry of Fisheries, Animal Husbandry and Dairying, Government of India (Registration No and Date-9/1999/CPCSEA). Male and female offspring of Balb/c mice bred in the institutional animal house were used for the study. All animals were housed under SPF conditions. They were kept in individually ventilated cages (IVC) at 24±2°C on a 12h light/dark cycle with ad libitum access to food and water. Animal handling and experiments were conducted in accordance with the institutional guidelines.

### Decision letter and Author response
Decision letter https://doi.org/10.7554/eLife.77968.sa1
Author response https://doi.org/10.7554/eLife.77968.sa2

---

## Additional files

### Supplementary files
• Supplementary file 1. List of differentially expressed genes (DEGs) in Hypo and PFC of Males and Females.

• Supplementary file 2. ANOVA Tables of *Figure 3*, *Figure 4* and *Figure 7*.

• Supplementary file 3. Primers used for qRT-PCR and MeDIP-qPCR.

• Transparent reporting form

### Data availability
RNA sequencing data have been deposited in GEO under accession code GSE199844. All data generated or analyzed during this study are included in the manuscript and supplementary files. Source data files have been provided for Fig. 1C, Fig. 3D, Fig. 4B-4J, Fig. 4P, Fig.5B-5D, Fig.6B-6E, Fig.7 and Fig. 4-figure supplement 1.

The following dataset was generated:

| Author(s) | Year | Dataset title | Dataset URL | Database and Identifier |
|---|---|---|---|---|
| Konar A | 2022 | Early life stressful experiences escalate aggressive behavior in adulthood via changes in transthyretin expression and function | http://www.ncbi.nlm.nih.gov/geo/query/acc.cgi?acc=GSE199844 | NCBI Gene Expression Omnibus, GSE199844 |

---

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
