## [Editor Report]

The adolescent phase of life, particularly that surrounding puberty, is a sensitive period for brain development such that adverse experiences during that time have enduring negative impacts but the mechanisms for how this occurs are largely unknown. This important study provides convincing evidence for an unexpected role for the thyroid hormone transporter, transthyretin, which shows region-specific changes in expression following peri-pubertal stress and increased aggression in males, but not females. Mimicking the changes in transthyretin expression induced by stress also increases aggressive behavior in adult males, suggesting a causal connection between changes in thyroid hormone signaling and the behavioral changes induced by stress around puberty.

---

## [Decision Letter]

**Decision letter after peer review:**

Thank you for submitting your article "Early life trauma leads to escalated aggressive behavior and its inheritance by impairing thyroid hormone availability in brain" for consideration by *eLife*. Your article has been reviewed by 3 peer reviewers, one of whom is a member of our Board of Reviewing Editors, and the evaluation has been overseen by Catherine Dulac as the Senior Editor. The following individual involved in review of your submission has agreed to reveal their identity: Petr N Menshanov (Reviewer #2).

Essential revisions:

1) The role of the decreased hypothalamic T3 availability in the development of aggression is not causally established. In the current form of the paper, it is not clear whether the decrease of TTR expression causes the aggression via the regulation of hypothalamic T3 availability or the effect of altered TTR expression on the aggression and T3 availability are independent. Therefore, the authors should either focus on the role of TTR or they should provide additional evidence that lack of hypothalamic T3 is the cause of the development of aggression.

2) The studies on epigenetic mediated inheritance of the aggression phenotype are not well supported and should be removed. The data on methylation of the TTR promoter are an important contribution and should remain but should not be considered as a diagnostic for aggressive behavior or the basis for trans-generational inheritance.

3) Statistical analyses should be redone with factorial analyses to make the proper comparisons between sexes, stress condition and brain region.

4) RNA-Seq data should include greater in depth analyses.

5) The images of immunohistochemistry staining are not of sufficient quality and should be replaced.

*Reviewer #1 (Recommendations for the authors):*

Deiodinase 3 (Dio3) plays critical role in the regulation of hypothalamic T3 availability, therefore, the D3 expression should also be determined along the other TH regulated genes.

According to my knowledge, there is no reliable antibody against dio2. This fact highly questions the validity of the dio2 ELISA data. The only accepted method for the determination of DIO2 protein level is the DIO2 enzyme assay (PMID: 24001133).

The quality of the images illustrating immunocytochemistry is very weak. Better images would be necessary. In addition, it should be described in more details where the TTR-immunoreactive cells were observed in the hypothalamus. In addition, colocalization study should be performed at least with neuronal, glial and endothelial markers to determine what kind of cells express TTR in the hypothalamus.

*Reviewer #2 (Recommendations for the authors):*

Suggestions for improvement

1.1. In the present form submitted to GEO database, the transcriptomic data on the levels of individual transcripts are reported in aggregate per each experimental group, with no possibility to estimate variance between individual biological replicates. Reporting raw transcriptomic data on the transcript levels in each biological replicate will be beneficial for the readers interested in reanalyses of the data.

It is indispensable to report key elements of the transcriptomic analysis in full. Please update the GEO data with a single file on all levels of individual transcripts in each individual biological replicate.

1.2. The personal experience of the reviewer evidences that old Cufflink-Cuffdiff Tuxedo bioinformatic pipeline for RNA-Seqs (done after Tophat or STAR aligners), if applied properly, is highly sensitive to individual mRNA levels with minor shares. Nevertheless, in the present form of the description of bioinformatics pipeline in the "Methods" section, with missed basic options applied by the authors for Tophat, Cufflink, Cuffdiff utilits, it is hard to estimate the validity of the transcriptomic analyses done by the authors. Please consider improving the description of the bioinformatics methods applied.

1.3. Technical figures like volcano plots are important to control the quality of RNA-Seq, but are too uninformative to demonstrate the results of the experiment. Please consider illustrating the differences between PFC and HPT changes in DEGs on a single figure panel by depicting the parallel changes in the transcript levels identified in PFC (for example, x-axis) and HPT (for example, y-axis). At the same time, it is possible to move the volcano plots to the supplementary figures.

1.4. Preliminary analysis done by the reviewer by applying tissue cell deconvolution methods evidenced in favour of possible specific trends in cell numbers in the limbic brain regions studied. In particular, it cannot be excluded that the juvenile stress episodes suppressed microglia numbers in both the PFC and the HPT.

Please consider providing the data on tissue cell composition in RNA-Seq individual samples with respect to the experimental groups.

(For details, see Sutton et al. 2022 https://doi.org/10.1038/s41467-022.28655-4 or other).

Such an analysis might be illustrative that stress itself is necessary but not sufficient to induce changes in the aggressive behaviours in affected male mice demonstrated excessive violence.

1.5. Please consider additional analyse of the data on the individual transcripts levels reported in the RNA-Seq analysis in a way similar to 2-way or 3-way (when appropriate) factorial ANOVA, to identify possible additive and non-additive patterns of changes in the levels of transcripts.

1.6. It will be also interesting to know on cell specificity of DEGs identified in PFC. Please consider checking the prevalence of individual DEGs with the help of Allen brain transcriptomic atlas (cited in Yao et al.,2021) or another one.

The above mentioned atlas can be assessed by the following link (https://portal.brain-map.org/atlases-and-data/rnaseq)

1.7. Please consider redrawing all figure panels depicted as traditional bars with the (Median, IQR, SD) box plots with individual data dots depicted. In particular, it is possible to achieve this at ease with the JASP freeware (https://jasp-stats.org/)

1.8. It is arguable to use GAPDH as a reference for qPCR assay in the experiments with stress exposures, since GAPDH levels might be affected and even programmed by stress experienced by animals.

Please consider providing a rationale on the applying of GAPDH as an internal standard for mRNA levels instead of β-actin or other mRNAs conventional for stress studies.

1.9. For future studies.

In the absence of cross-fostering experimental schedule for F1 experiments, it is hard to delineate the origins of epigenetic changes identified in the F1 descendants, whether these changes were transmitted directly or indirectly, via mother's specific behaviours on the descendants.

Recommendations for improving the writing and presentation

2.1. The present description of animal procedures in the "Methods" section does not provide enough details on the environments, in which the experimental mice were grown up. Please provide all details on animal housing procedures that might be stressful (social isolation events, social crowding events, numbers of animals per cell etc)

2.2. The description of several methods must be improved to clarify details critical for data comprehension.

For example,

Please indicate the details of screening tests done with subjects (females and anesthetized intruder) that were attacked by mice with pathological aggressive behaviours.

The experimental schedules must be reported in a clear way also. In particular, a brief statement must be done on how the distinct populations of "adult control" and "PPS adult male" mice screened by the Authors were originated from. A similar clarification must be done for female mice groups also.

2.3. The initial two paragraphs in the "Introduction" section do not provide the linear story tale on violence, "escalated" violence and a difference of these two concepts of aggression. The logic of this section must be improved.

2.3a. In the introduction, it looks like that the authors consider an aggression trait as a behavioural continuum between "zero-level" aggression to appropriate violence, and then to "escalated" aggressive behaviour. This point of view is arguable since it cannot be excluded that aggression is a multidimensional trait. Please consider revising and clarifying.

2.3b. It is possible to criticize "escalated" aggressive behaviour as unproductive. However, please do not make generalized negative statements on the nature of general violence in the Introduction and in the Discussion. Such statements might blackmail protective types of aggression critical for survival in mammals and humans.

2.4. Please avoid to made generalized statements that "A can lead to B". Such statements with a strong modal verb "can" are highly misleading since the development of behavioural traits is not linear and depends on both genotype and environmental context often. Better to speak that "A might lead to B" under certain circumstances.

Several examples:

2.4a."Escalated aggressive behavior … can lead to antisocial and criminal activities" [P3-S1-LL1-2]. This is misleading for numerous animals and for specific types of aggressive behaviours in *H. sapiens*. Please consider revising.

2.4b. "…pathological aggression has emerged as a consequence of early life adversities…" [P3-S2.LL9-10]. This statement is misleading, as it blames unrightfully all children affected by harsh life. Please consider revising.

2.4c. "…we inferred that Ttr promoter methylation could serve as a predictor of … behavioral deficits." [P24-S1-LL14-15] – Better to talk about "possible behavioural deficits", not about "behavioural deficits".

The above-mentioned list of examples is not exhaustive.

*Reviewer #3 (Recommendations for the authors):*

1) Examination of some of the DEGs from the other pathways identified in the KEGG analysis would either strengthen the argument that changes are specific to the TH pathway or highlight that changes are more wide spread.

2) Some discussion of how a change in the amount of T3 and T4 could lead to aberrant aggression would enhance the manuscript.

3) Overall the sex difference, which is profound, is not given much attention. There are more DEGs identified in females subject to peripubertal stress than males, yet there is no change in aggressive behavior, so what does this tell us? Also, does testosterone play a role in the sex difference in both the transcriptome and the behavioral changes? If males were gonadectomized, would the same transcriptional profile be apparent and would the behavior also be there? Or would the two endpoints diverge, belying the noting that there is a casual connect between them.

4) The peri-pubertal stress was conducted on 7 random days from PN28 to PN42. The timing of puberty is different in males and females, being earlier in females. Where measures taken to determine the stage of puberty in each animal (i.e. vaginal opening, preputial separation)? Did the stress impact the timing of puberty?

5) The transgenerational assertions should either be dropped or the study carried out to the F2 generation.

6) How was the use of MeDIP specific to the promoter for Ttr?

7) What are the circulating androgen levels in the males from the various groups? Could the PP have altered the HPGA that then in turns alters behavior?

8) It does not seem appropriate to refer to "donut shaped cells".

9) Figure 4J – appears mislabeled, has Hypo twice and no PFC.

10) Figure 4L-P – why aren't the individual points plotted for the mRNA and protein.

11) Whenever both brain areas are considered the statistics should be 2-way ANOVA with brain region and treatment as factors?

12) The word "trauma" in the context used here connotes an emotional interpretation of stressful or fearful events. We do not know if the mice are experiencing trauma, instead we know they are being subject to fearful and stress-inducing experiences. It is suggested that the word trauma be removed throughout and replaced with more precise terminology.

[Editors' note: further revisions were suggested prior to acceptance, as described below.]

Thank you for resubmitting your work entitled "Pivotal role of TTR in early life stressful experiences induced escalated aggressive behavior" for further consideration by *eLife*. Your revised article has been evaluated by Catherine Dulac (Senior Editor) and a Reviewing Editor.

The manuscript has been improved but there are some remaining issues that need to be addressed, as outlined in detail below:

The following additional revisions are essential:

1) The conclusions on large sex differences between male and female hypothalamic transcriptomes are not supported by the data due to the problematic sample GSM5988437 (Female Hypo Experimental BiologicalReplicate1). This sample must be withdrawn from the analysis, and the male-female comparisons for hypothalamic transcriptomes must be re-estimated without this problematic sample. This will help to dismiss any spurious claims about the "augmented" male-female hypothalamic differences.

2) Data about the specificity of the TTR immunocytochemistry and D2 elisa would be absolutely necessary. The authors describe the changes of 2 TH transmitters and Dio2. Dio3 is at least an important regulator of TH availability in the brain as Dio2. So either data about Dio3 expression should be added or data about the expression of TH transporters and Dio2 should be removed.

3) Determination of the cell types expressing TTR should be very fast and easy with double-labeling immunocytochemistry and would increase the value of the paper, however, this is not absolutely necessary to support the conclusions of the paper.

*Reviewer #1 (Recommendations for the authors):*

The manuscript was highly improved, but there are some points that need further change.

The immunocytochemical images of Figure 3 were highly improved. On E, the TTR immunoreactivity seems to be neuronal, while the TTR seems to be present in perivascular localization suggesting endothelial or astrocytic expression. The cell type expressing TTR in the studied brain regions should be determined using double-labeling immunofluorescence.

The specificity of the TTR immunostaining should be proven.

It should be described which hypothalamic region is shown in the images.

The red immunofluorescent signal is visible inside the ventricle on F. This questions the specificity of the staining.

The length of the scale bars is hardly visible. In addition, while the magnification of the left and right panels are obviously different, the scale bars are the same on all images. This should be corrected.

As the balance of DIO2 and DIO3 determines the TH availability, in addition to the expression level of transporters and DIO2, the expression of DIO3 should be also presented.

As there is no specific DIO2 antibody, the validity of the DIO2 protein data is highly questionable. Therefore, data about the specificity of the assay should be provided.

Dio2 is negatively regulated by TH in most parts of the brain, but changes in TH level have no effect on Dio2 activity in the hypothalamus suggesting that the observed changes of Dio2 are not compensatory.

*Reviewer #2 (Recommendations for the authors):*

1. It is still unclear whether Tophat, Cufflinks, Cuffmerge, and Cuffdiff tools were applied with the default settings or specific settings.

Please indicate clearly the settings applied for Tophat, Cuffdiff, and other bioinformatic tools used, within the "RNA-sequencing" subsection of the "Materials and methods" section.

2. Individual data provided for RNA-seq evidenced that Cuffdiff2 "misbehaved" during bioinformatics analysis while estimating FPKM levels for some abundant transcripts and filtering biological noise. As a result, Cuffdiff2 nullifies FPKM levels for some abundant transcripts like mitochondrial mt-Nd6 non-uniformly, only in specific samples. This issue should not be considered as a large problem, as Cuffdiff2's misbehaviour in such situations is already known.

To deal with this issue, the authors must include in the "Discussion" section a short sentence on this possible limitation of the study arising when comparing levels of highly abundant transcripts caused by Cuffdiff2 due to normalizations enacted.

3. The sample GSM5988437 (Female Hypo Experimental BiologicalReplicate1) failed to pass the Replicate concordance test (R=0.59-0.63), while all other replicates demonstrated R>0.8. To understand the Replicate concordance test, please consult with the ENCODE guidelines on RNA-Seq (https://www.encodeproject.org/about/experiment-guidelines/).

Since all reads were found to be above phred score 22 with no adapter contamination or over-represented sequences P46-LL882.883. , the FPKM levels in the sample GSM5988437 (Female Hypo Experimental BiologicalReplicate1) were nullified somehow (by Cuffdiff2 or in another way) for a large number of transcripts with 1.to-10 FPKM levels, as compared to other samples.

This might result in a spurious identification of DEGs in the female hypothalami of exposed animals and in a spurious identification of sex differences between male and female hypothalamic DEGs (please see Supplementary File 1 with FPKMs, and the Figures 1D, 2A, and 2D).

To deal with these complications,

3a. The authors must repeat the concordance test for their FPKM estimates in the individual samples, providing the results of the concordance test in the supplement Table in accordance with the ENCODE guidelines.

3b. It is necessary to repeat a Cuffdiff analysis for hypothalamic RNA-seq samples without the affected replicate GSM5988437 (Female Hypo Experimental BiologicalReplicate1), to ensure unbiased data for the Figure 2A, Figure 2D, DEG lists and the "Transcriptome analyses identify brain region-specific gene signatures…" subsection.

3c. The subsection "Transcriptome analyses identify brain region-specific gene signatures…" must be improved after a Cuffdiff-mediated reanalysis of the RNA-Seq data.

4. ANOVA results gained by the authors are an important addon improving the text comprehensibility. Nevertheless, there is no need to incorporate ANOVA Tables into the manuscript text. Please move these Tables to Supplement.

5. Please identify units of measurements for X- and Y-axes on the subpanels of Figures1D and 2A.

[Editors' note: further revisions were suggested prior to acceptance, as described below.]

Thank you for resubmitting your work entitled "Pivotal role of TTR in early life stressful experiences induced escalated aggressive behavior" for further consideration by *eLife*. Your revised article has been evaluated by Catherine Dulac (Senior Editor) and a Reviewing Editor.

The manuscript has been improved but there are some remaining issues that need to be addressed, as outlined below:

Thank you for the additional revisions to your manuscript. Upon further examination, it appears that some of the wording continues to be imprecise and is at times inappropriate in over-interpreting the significance of the findings to human behavior. Some specific suggestions are provided below but it would be useful to revisit the entire manuscript with an eye towards assuring there are no hidden biases and that are statements are both factually and grammatically correct.

Specific Comments:

1) The change in the title now makes it both awkward and inaccessible to the general reader. One suggestion is: "Early life stressful experiences escalate adult mouse aggressive behavior via changes in transthyretin expression and function" or some variation thereof.

2) There are many examples of statements regarding violence and criminality that are inappropriate or incorrectly attribute to causality. For instance, the second sentence of the Abstract states "Early life stressful experiences triggers adulthood violence and criminality". This suggests that anyone that experiences early life stress will grow up to be a violent criminal. It is critically important not to make blanket statements that can be misinterpreted as sound science.

3) It is unclear how there can be an "escalated aggressive phenotype", a phenotype cannot be escalated, perhaps instead state "enhanced aggressive phenotype" or "resulted in escalated aggressive behavior".

4) Second sentence of the Introduction – "Such aberrant behavioral patterns are also manifested in patients of multiple psychiatric disorders including schizophrenia and bipolar disorder (2, 3) necessitating the identification of predisposing factors and early intervention strategies." – this seems strongly stigmatizing of individuals with mental illness, the vast majority of whom are not violent. It also is not important to the current findings and I would recommend removing such references to mental illness entirely.

5) Introduction – "Brain region-specific long-term changes in Ttr gene expression and thyroid hormone (TH) availability was evident in PPS induced escalated aggressive male mice, circulating TH being unaltered". – should read "were evident".

6) Introduction – "….it is extremely important to understand the biological culprits underlying brutal shift of normal adaptive aggression to escalated and pathological form." – the words "culprits" and "brutal" are emotionally laden terms that are inappropriate when discussing research findings.

7) Introduction – "We selected the extreme phenotypes for better understanding of the behavior observed in human violent offenders and psychopathy" – it is important to limit the conclusions of your study to what you observed which was changes in mouse behavior. Given the enormous complexity and multifactorial nature of violence in humans, it behooves you to not try and make direct connections between your studies and humans in the absence of any evidence that similar mechanisms are at play in human violent offenders.

---

## [Author Response]

Essential revisions:1) The role of the decreased hypothalamic T3 availability in the development of aggression is not causally established. In the current form of the paper, it is not clear whether the decrease of TTR expression causes the aggression via the regulation of hypothalamic T3 availability or the effect of altered TTR expression on the aggression and T3 availability are independent. Therefore, the authors should either focus on the role of TTR or they should provide additional evidence that lack of hypothalamic T3 is the cause of the development of aggression.

As suggested, the focus of the paper has been shifted to the role of TTR in aggression and accordingly title has been modified to “Pivotal role of TTR in early life stressful experiences induced escalated aggressive behavior” in the revised version of the manuscript.

2) The studies on epigenetic mediated inheritance of the aggression phenotype are not well supported and should be removed. The data on methylation of the TTR promoter are an important contribution and should remain but should not be considered as a diagnostic for aggressive behavior or the basis for trans-generational inheritance.

As suggested, studies on epigenetic mediated inheritance of the aggression phenotype that is Ttr promoter methylation changes in F1 generation (Figure 7B) has been removed. Behavioural and other molecular studies (Ttr and thyroid hormone gene expression) in F1 generation has been kept as an important observation (Figure 6) without claiming anything on epigenetic inheritance. However, if the reviewers wish the behavioural and molecular studies in F1 generation also to be removed, we are open to do the modifications.

3) Statistical analyses should be redone with factorial analyses to make the proper comparisons between sexes, stress condition and brain region.

Two way ANOVA with stress condition and brain region as factors (Revised Figure 3D, Figure 4F-4P and Figure 7) and three way ANOVA with sex, stress condition and brain region as factors (Revised Figure 3B and Figure 3C) has been performed and described in the result and figure legend section of the revised manuscript. ANOVA summary tables (Table 1 to Table 15) have also been incorporated in the main manuscript and details of ANOVA analyses for all the above mentioned figures have been given in Supplementary File 2.

4) RNA-Seq data should include greater in depth analyses.

As suggested, RNA-Seq data has been analysed in depth, addressing the following points which are also incorporated in revised manuscript.

GEO data has been updated with FPKM values of individual transcripts in individual biological replicate of control and experimental groups. Private token for access of reviewers has been provided.Single figure panel depicting parallel changes in transcripts in PFC and Hypo has been included (Figure 1D and 2A)Cell type specificity analyses of Male DEGs by Barres Lab database (brainrnaseq.org) and Allen brain transcriptomic atlas (https://portal.brain-map.org/atlases-and-data/rnaseq) has been included as Figure 1—figure supplement 2.10.

5) The images of immunohistochemistry staining are not of sufficient quality and should be replaced.

As suggested, IHC images of better quality and clarity has been included in the revised manuscript (Figure 3E-3H)

Reviewer #1 (Recommendations for the authors):Deiodinase 3 (Dio3) plays critical role in the regulation of hypothalamic T3 availability, therefore, the D3 expression should also be determined along the other TH regulated genes.According to my knowledge, there is no reliable antibody against dio2. This fact highly questions the validity of the dio2 ELISA data. The only accepted method for the determination of DIO2 protein level is the DIO2 enzyme assay (PMID: 24001133).

It is very true that Deiodinase 3 (Dio3) plays crucial role in the regulation of hypothalamic T3 availability and therefore, determining the D3 expression would provide further insights into TH signalling in brain. In view of the comment 1 under “Essential revision” section, we have now shifted the focus of the manuscript on TTR and limited our claims on TH availability. However, authors appreciate suggestion of the reviewer and future studies will include both Dio2 and Dio3 with more advanced methods in the context of aggressive behaviour.

The quality of the images illustrating immunocytochemistry is very weak. Better images would be necessary. In addition, it should be described in more details where the TTR-immunoreactive cells were observed in the hypothalamus. In addition, colocalization study should be performed at least with neuronal, glial and endothelial markers to determine what kind of cells express TTR in the hypothalamus.

As suggested, IHC images of better quality and clarity has been included in the revised manuscript (Figure 3E-H). Hypothalamic regions in which TTR immunoreactive cells were observed, have been described in result section of Figure 3E.

Cells expressing TTR in hypothalamus have been highlighted in Figure 3E. Barres Lab database (brainrnaseq.org) analyses revealed presence of Ttr mRNA in different mouse brain cell types (endothelial cells> neurons> microglia> astrocytes> oligodendrocytyes). We have given this information in Figure 1—figure supplement 4. However, co-localization studies with appropriate neuronal, glial and endothelial markers for TTR protein are warranted to confirm the cell type specificity.

Reviewer #2 (Recommendations for the authors):Suggestions for improvement1.1. In the present form submitted to GEO database, the transcriptomic data on the levels of individual transcripts are reported in aggregate per each experimental group, with no possibility to estimate variance between individual biological replicates. Reporting raw transcriptomic data on the transcript levels in each biological replicate will be beneficial for the readers interested in reanalyses of the data.It is indispensable to report key elements of the transcriptomic analysis in full. Please update the GEO data with a single file on all levels of individual transcripts in each individual biological replicate.

As suggested GEO data has been updated with FPKM values of all transcripts in individual biological replicate of all the control and experimental groups.

In addition, Supplementary File 1 containing FPKM values of individual transcripts in individual biological replicate of control and experimental groups has been incorporated in the revised manuscript.

1.2. The personal experience of the reviewer evidences that old Cufflink-Cuffdiff Tuxedo bioinformatic pipeline for RNA-Seqs (done after Tophat or STAR aligners), if applied properly, is highly sensitive to individual mRNA levels with minor shares. Nevertheless, in the present form of the description of bioinformatics pipeline in the "Methods" section, with missed basic options applied by the authors for Tophat, Cufflink, Cuffdiff utilits, it is hard to estimate the validity of the transcriptomic analyses done by the authors. Please consider improving the description of the bioinformatics methods applied.

As suggested, description of the bioinformatics methods applied in RNA seq data analyses has been improved and incorporated in Materials and methods section of the revised manuscript.

1.3. Technical figures like volcano plots are important to control the quality of RNA-Seq, but are too uninformative to demonstrate the results of the experiment. Please consider illustrating the differences between PFC and HPT changes in DEGs on a single figure panel by depicting the parallel changes in the transcript levels identified in PFC (for example, x-axis) and HPT (for example, y-axis). At the same time, it is possible to move the volcano plots to the supplementary figures.

As suggested a single figure panel (Figure 1D) depicting the parallel changes in transcript levels (log2 Fold change) identified in PFC (x-axis) and Hypo (y-axis) has been incorporated and volcano plots has been moved to Figure 1—figure supplement 1 (Male DEGs) and Figure 2—figure supplement 1 (Female DEGs) of the revised manuscript.

1.4. Preliminary analysis done by the reviewer by applying tissue cell deconvolution methods evidenced in favour of possible specific trends in cell numbers in the limbic brain regions studied. In particular, it cannot be excluded that the juvenile stress episodes suppressed microglia numbers in both the PFC and the HPT.Please consider providing the data on tissue cell composition in RNA-Seq individual samples with respect to the experimental groups.(For details, see Sutton et al. 2022 https://doi.org/10.1038/s41467-022.28655-4 or other)Such an analysis might be illustrative that stress itself is necessary but not sufficient to induce changes in the aggressive behaviours in affected male mice demonstrated excessive violence.

Applying tissue cell deconvolution methods in our RNA-seq data is indeed a very good suggestion and will definitely provide deeper insights. Due to technical constraints, we could not perform the exact analyses asked for, though we have provided the cell type data for PFC and Hypo DEGs in Figure 1—figure supplement 2-10 using other databases Barres Lab database (brainrnaseq.org) and Allen Brain Map website (https://celltypes.brain-map.org/).

1.5. Please consider additional analyse of the data on the individual transcripts levels reported in the RNA-Seq analysis in a way similar to 2-way or 3-way (when appropriate) factorial ANOVA, to identify possible additive and non-additive patterns of changes in the levels of transcripts.

The RNA sequencing experiment was performed using three replicates for each condition. All these replicates were then used for analysis using the Tuxedo pipeline which uses Cuffdiff2 to determine differentially expressed genes.

Cuffdiff2 combines a β distribution model to account for the count uncertainty arising from the high sequencing depth while it uses a negative binomial distribution to take care of the over dispersion problem that may exist during sequencing. The algorithm counts the fragments aligned per kb of exon per million reads which is then used to calculate p value. Moreover, it performs a Benjamini-Hochberg correction on the p value to account for false-discovery rate. After the whole procedure, the corrected p value (also labelled as q value in the output) is a statistically trusted significance measure. Therefore, ANOVA may not be required on this value as it would not account for FDR. Further, it will not be able to account for biological noise which do not show up in the FPKM values of individual replicates. The noise is accounted for by analysis of raw alignment data which Cuffdiff2 performs. The algorithm is used widely in the field and currently has more than 3000 citations.

The authors completely agree with reviewers that ANOVA is essential when analyzing qRT-PCR data and therefore, it has been included in relevant graphs of the revised manuscript.

1.6. It will be also interesting to know on cell specificity of DEGs identified in PFC. Please consider checking the prevalence of individual DEGs with the help of Allen brain transcriptomic atlas (cited in Yao et al.,2021) or another one.The above mentioned atlas can be assessed by the following link (https://portal.brain-map.org/atlases-and-data/rnaseq)

Investigating cell type specificity of DEGs is indeed a very good idea. However, not all genes from our RNA seq analysis can be searched in publicly available data since we performed Total RNA sequencing while majority of researchers only perform polyA+ RNA sequencing. We have extracted the data on cell type of top ranking PFC and Hypo DEGs from Barres Lab database (brainrnaseq.org) which was made by bulk sequencing of separated cells. Also, we checked cell type specificity of some top ranking PFC DEGs in Allen Brain Map website (https://celltypes.brain-map.org/) and generated heat map. The figures are provided as Figure 1 —figure supplement 2.10 and mentioned in result section.

1.7. Please consider redrawing all figure panels depicted as traditional bars with the (Median, IQR, SD) box plots with individual data dots depicted. In particular, it is possible to achieve this at ease with the JASP freeware (https://jasp-stats.org/)

As suggested, individual data points have been depicted in figure panels.

1.8. It is arguable to use GAPDH as a reference for qPCR assay in the experiments with stress exposures, since GAPDH levels might be affected and even programmed by stress experienced by animals.Please consider providing a rationale on the applying of GAPDH as an internal standard for mRNA levels instead of β-actin or other mRNAs conventional for stress studies.

We did not get any change in GAPDH mRNA levels in our transcriptome data and as per previous literature related to our experimental regime and other stress exposure studies (some references given below) in rodents, GAPDH is widely being used as an internal control. Therefore, we used GAPDH as an internal standard for mRNA levels. Ttr mRNA expression was also normalized with 18s rRNA but the results did not vary (data not shown).

References

Kathleen E. Morrison et al. (2016) Peripubertal Stress With Social Support Promotes Resilience in the Face of Aging, Endocrinology. 2016 May; 157(5): 2002–2014.

Jordi Tomas‐Roig et al. (2018). Effects of repeated long‐term psychosocial stress and acute cannabinoid exposure on mouse corticostriatal circuitries: Implications for neuropsychiatric disorders. CNS Neurosci Ther 24(6): 528–538.

Du, P et al. (2020). Chronic stress promotes EMT-mediated metastasis through activation of STAT3 signaling pathway by miR-337-3p in breast cancer. Cell Death Dis 11, 761.

1.9. For future studies.In the absence of cross-fostering experimental schedule for F1 experiments, it is hard to delineate the origins of epigenetic changes identified in the F1 descendants, whether these changes were transmitted directly or indirectly, via mother's specific behaviours on the descendants.

We have limited our claims in the revised version of the manuscript and studies on epigenetic mediated inheritance of the aggression phenotype has been removed. However, further studies with cross-fostering experimental schedule till F2 generation are in progress to elucidate the molecular basis of inheritance in behaviour.

Recommendations for improving the writing and presentation2.1. The present description of animal procedures in the "Methods" section does not provide enough details on the environments, in which the experimental mice were grown up. Please provide all details on animal housing procedures that might be stressful (social isolation events, social crowding events, numbers of animals per cell etc)

ARRIVE guidelines for animal research has already been uploaded as supporting document. Also, we have included details in the “Methods” section describing animal housing environments.

2.2. The description of several methods must be improved to clarify details critical for data comprehension.For example,Please indicate the details of screening tests done with subjects (females and anesthetized intruder) that were attacked by mice with pathological aggressive behaviours.The experimental schedules must be reported in a clear way also. In particular, a brief statement must be done on how the distinct populations of "adult control" and "PPS adult male" mice screened by the Authors were originated from. A similar clarification must be done for female mice groups also.

As suggested, methods have been described in detail for better understanding of the readers.

2.3. The initial two paragraphs in the "Introduction" section do not provide the linear story tale on violence, "escalated" violence and a difference of these two concepts of aggression. The logic of this section must be improved.2.3a. In the introduction, it looks like that the authors consider an aggression trait as a behavioural continuum between "zero-level" aggression to appropriate violence, and then to "escalated" aggressive behaviour. This point of view is arguable since it cannot be excluded that aggression is a multidimensional trait. Please consider revising and clarifying.2.3b. It is possible to criticize "escalated" aggressive behaviour as unproductive. However, please do not make generalized negative statements on the nature of general violence in the Introduction and in the Discussion. Such statements might blackmail protective types of aggression critical for survival in mammals and humans.

Introduction and Discussion section has been revised in light of the above comments.

2.4. Please avoid to made generalized statements that "A can lead to B". Such statements with a strong modal verb "can" are highly misleading since the development of behavioural traits is not linear and depends on both genotype and environmental context often. Better to speak that "A might lead to B" under certain circumstances.Several examples:2.4a. "Escalated aggressive behavior … can lead to antisocial and criminal activities" [P3-S1-LL1-2]. This is misleading for numerous animals and for specific types of aggressive behaviours in *H. sapiens*. Please consider revising.

The statement has been revised in light of the above comment.

2.4b. "…pathological aggression has emerged as a consequence of early life adversities…" [P3-S2-LL9-10]. This statement is misleading, as it blames unrightfully all children affected by harsh life. Please consider revising.

The statement has been revised in light of the above comment.

2.4c. "…we inferred that Ttr promoter methylation could serve as a predictor of … behavioral deficits." [P24-S1-LL14-15]. – Better to talk about "possible behavioural deficits", not about "behavioural deficits".

As suggested in the essential revision section, the entire statement has now been removed in text of revised manuscript.

The above-mentioned list of examples is not exhaustive.Reviewer #3 (Recommendations for the authors):1) Examination of some of the DEGs from the other pathways identified in the KEGG analysis would either strengthen the argument that changes are specific to the TH pathway or highlight that changes are more wide spread.

Considering the key role of TTR in TH transport and KEGG analyses showing TH as the topmost ranking pathway, we explored TH in detail. However, DEGs from other pathways are worth exploring in the future. It is also possible that TH directly or indirectly influences the other pathways as well.

2) Some discussion of how a change in the amount of T3 and T4 could lead to aberrant aggression would enhance the manuscript.

Considering the comments mentioned in the “Essential revision” section, the focus of the manuscript has now been shifted to TTR and it is important to mention that further studies are necessary to identify the precise molecular pathway responsible for aberrant aggression. Therefore detailed discussion on role of T3 and T4 has been not been incorporated in the revised manuscript.

3) Overall the sex difference, which is profound, is not given much attention. There are more DEGs identified in females subject to peripubertal stress than males, yet there is no change in aggressive behavior, so what does this tell us? Also, does testosterone play a role in the sex difference in both the transcriptome and the behavioral changes? If males were gonadectomized, would the same transcriptional profile be apparent and would the behavior also be there? Or would the two endpoints diverge, belying the noting that there is a casual connect between them.

The authors really appreciate the suggestion to investigate sex differences in detail, though this was not the prime focus of this manuscript. However, work with similar objectives suggested by reviewer is in progress.

4) The peri-pubertal stress was conducted on 7 random days from PN28 to PN42. The timing of puberty is different in males and females, being earlier in females. Where measures taken to determine the stage of puberty in each animal (i.e. vaginal opening, preputial separation)? Did the stress impact the timing of puberty?

The timing of peripubertal stress exposure from PN28-PN42 is well established both for male and female mice {Marquez et al. (2013) Transl Psychiatry,15;3(1):e216; Kathleen E. Morrison et al. (2016) Endocrinology.157(5): 2002–2014;Morató L et al. (2022) Sci Adv, 8(9):eabj9109.}

However, we also assessed pubertal onset in males through preputial separation and in females by observing vaginal opening and determining start of estrous (sexual) by crystal violet staining based vaginal cytology. We did not observe any conspicuous changes in the above mentioned visible signs of pubertal onset in both sexes, though it is indeed a very interesting point and worth exploring in detail in future studies.

5) The transgenerational assertions should either be dropped or the study carried out to the F2 generation.

We have not made any transgenerational assertions, rather referred to the observations as “intergenerational inheritance” as we conducted experiments only in F1 generation. However, studies related to epigenetic mediated inheritance of aggressive phenotype has been removed in the revised version of manuscript

6) How was the use of MeDIP specific to the promoter for Ttr?

Ttr proximal promoter (-184 to -33 bp from TSS) was ampliﬁed with specific primers (Supplementary File 3) generating a 151 bp product in MedIP-qPCR and this has been mentioned in methods section.

7) What are the circulating androgen levels in the males from the various groups? Could the PP have altered the HPGA that then in turns alters behavior?

The primary objective of the study was to identify brain region specific transcriptional responses and as we did not find androgen signalling genes in our top DEGs, we did not dig down further. However, high testosterone and testosterone/cortisol ratio being considered important for aggression, circulating androgens and PPS triggered changes in HPGA axis is worth exploring in future studies.

8) It does not seem appropriate to refer to "donut shaped cells".

The term "donut shaped cells" has been removed from the text.

9) Figure 4J – appears mislabeled, has Hypo twice and no PFC.

Mislabelling has been corrected in revised Figure 4J.

10) Figure 4L-P – why aren't the individual points plotted for the mRNA and protein.

As suggested individual points have been plotted for the mRNA and protein in revised Figure 4L-P.

11) Whenever both brain areas are considered the statistics should be 2-way ANOVA with brain region and treatment as factors?

As suggested 2-way ANOVA has been performed with brain region and treatment as factors (Revised Figure 3D, Figure 4F-4P and Figure 7) and 3-way ANOVA has been performed with sex, brain region and treatment as factors (Revised Figure 3B and Figure 3C).

12) The word "trauma" in the context used here connotes an emotional interpretation of stressful or fearful events. We do not know if the mice are experiencing trauma, instead we know they are being subject to fearful and stress-inducing experiences. It is suggested that the word trauma be removed throughout and replaced with more precise terminology.

As suggested the word “trauma” has been replaced with “stressful experiences” throughout the text of the revised manuscript.

[Editors' note: further revisions were suggested prior to acceptance, as described below.]

The manuscript has been improved but there are some remaining issues that need to be addressed, as outlined in detail below:The following additional revisions are essential:1) The conclusions on large sex differences between male and female hypothalamic transcriptomes are not supported by the data due to the problematic sample GSM5988437 (Female Hypo Experimental BiologicalReplicate1). This sample must be withdrawn from the analysis, and the male-female comparisons for hypothalamic transcriptomes must be re-estimated without this problematic sample. This will help to dismiss any spurious claims about the "augmented" male-female hypothalamic differences.

We really appreciate the suggestion of reviewers and thus we have withdrawn Female Hypo Experimental Biological Replicate1 sample, redone the analysis and revised the relevant figure (Figure 2) in the manuscript. We have included the updated list of DEGs in females along with earlier male DEGs in supplementary file 1. Also, we mentioned in the RNA seq method section that replicates that passed the concordance test (R> 0.8) were included for analysis of DEGs.

2) Data about the specificity of the TTR immunocytochemistry and D2 elisa would be absolutely necessary. The authors describe the changes of 2 TH transmitters and Dio2. Dio3 is at least an important regulator of TH availability in the brain as Dio2. So either data about Dio3 expression should be added or data about the expression of TH transporters and Dio2 should be removed.

No primary TTR antibody control was used to determine the specificity of TTR immunofluorescence and included as supplementary data (Figure 3—figure supplement 1) Moreover, the same TTR primary antibody has been used for TTR western blotting that gave the precise molecular weight band further confirming the specificity.

Considering the need for timely publication of our main finding that TTR is involved in stress driven aggression, we are removing the data about the expression of TH transporters and Dio2 and accordingly revised figures have been included (Figure 4. Figure 5 and Figure 6).

However, future studies focussing on role of brain TH availability in aggression and more importantly determining Dio2 and Dio3 levels with more valid and advanced methods is definitely on priority.

3) Determination of the cell types expressing TTR should be very fast and easy with double-labeling immunocytochemistry and would increase the value of the paper, however, this is not absolutely necessary to support the conclusions of the paper.

We absolutely agree that determination of the cell types expressing TTR would increase the value of the paper. However, we have shifted our laboratory to a new place and the project involving this study is also completed. Therefore, doing the co-localization experiments will take more time than anticipated. In our future planned experiments co-localization of TTR protein with appropriate neuronal, glial and endothelial markers in mouse brain is definitely on the priority list.

[Editors' note: further revisions were suggested prior to acceptance, as described below.]

The manuscript has been improved but there are some remaining issues that need to be addressed, as outlined below:Thank you for the additional revisions to your manuscript. Upon further examination, it appears that some of the wording continues to be imprecise and is at times inappropriate in over-interpreting the significance of the findings to human behavior. Some specific suggestions are provided below but it would be useful to revisit the entire manuscript with an eye towards assuring there are no hidden biases and that are statements are both factually and grammatically correct.Specific Comments:1) The change in the title now makes it both awkward and inaccessible to the general reader. One suggestion is: "Early life stressful experiences escalate adult mouse aggressive behavior via changes in transthyretin expression and function" or some variation thereof.

As suggested the title has now been modified to “Early life stressful experiences escalate aggressive behavior in adulthood via changes in transthyretin expression and function”

2) There are many examples of statements regarding violence and criminality that are inappropriate or incorrectly attribute to causality. For instance, the second sentence of the Abstract states "Early life stressful experiences triggers adulthood violence and criminality". This suggests that anyone that experiences early life stress will grow up to be a violent criminal. It is critically important not to make blanket statements that can be misinterpreted as sound science.

As suggested the statement has been modified to “. Early life stressful experiences might increase the risk of developing pathological aggressive behavior in adulthood

3) It is unclear how there can be an "escalated aggressive phenotype", a phenotype cannot be escalated, perhaps instead state "enhanced aggressive phenotype" or "resulted in escalated aggressive behavior".

As suggested the above mentioned phrase of “escalated aggressive phenotype in the abstract has been changed to “resulted in escalated aggressive behavior”. In addition escalated aggressive phenotype has been changed to escalated aggressive behavior throughout the text.

4) Second sentence of the Introduction – "Such aberrant behavioral patterns are also manifested in patients of multiple psychiatric disorders including schizophrenia and bipolar disorder (2, 3) necessitating the identification of predisposing factors and early intervention strategies." – this seems strongly stigmatizing of individuals with mental illness, the vast majority of whom are not violent. It also is not important to the current findings and I would recommend removing such references to mental illness entirely.

As suggested the above references have been removed.

5) Introduction – "Brain region-specific long-term changes in Ttr gene expression and thyroid hormone (TH) availability was evident in PPS induced escalated aggressive male mice, circulating TH being unaltered". – should read "were evident".

As suggested the statement has been corrected to Brain region-specific long-term changes in Ttr gene expression and thyroid hormone (TH) availability were evident in PPS induced escalated aggressive male mice, circulating TH being unaltered.

6) Introduction – "….it is extremely important to understand the biological culprits underlying brutal shift of normal adaptive aggression to escalated and pathological form." – the words "culprits" and "brutal" are emotionally laden terms that are inappropriate when discussing research findings.

As suggested the statement has been modified as “it is extremely important to understand the biological factors contributing to shift of normal adaptive aggression to escalated and pathological form"

7) Introduction – "We selected the extreme phenotypes for better understanding of the behavior observed in human violent offenders and psychopathy" – it is important to limit the conclusions of your study to what you observed which was changes in mouse behavior. Given the enormous complexity and multifactorial nature of violence in humans, it behooves you to not try and make direct connections between your studies and humans in the absence of any evidence that similar mechanisms are at play in human violent offenders.

The comment is highly appreciated and accordingly all the sentences including “We selected the extreme phenotypes for better understanding of the behavior observed in human violent offenders and psychopathy” making direct connections with human violence behavior have been removed.